# Parasitism causes changes in caterpillar odours and associated bacterial communities with consequences for host-location by a hyperparasitoid

**Mitchel E. Bourne**[1]☺*, **Gabriele Gloder**[2,3]☺, **Berhane T. Weldegergis**[1], **Marijn Slingerland**[1], **Andrea Ceribelli**[1], **Sam Crauwels**[2,3], **Bart Lievens**[2,3], **Hans Jacquemyn**[3,4], **Marcel Dicke**[1], **Erik H. Poelman**[1]*

1 Laboratory of Entomology, Wageningen University & Research, Wageningen, The Netherlands, 2 CMPG Laboratory for Process Microbial Ecology and Bioinspirational Management (PME&BIM), Department M2S, KU Leuven, Leuven, Belgium, 3 Leuven Plant Institute (LPI), KU Leuven, Leuven, Belgium, 4 Laboratory of Plant Conservation and Population Biology, Biology Department, KU Leuven, Leuven, Belgium

☺ These authors contributed equally to this work.
* mitchel.bourne@wur.nl (MEB); erik.poelman@wur.nl (EHP)

**Data Availability Statement:** The sequences obtained in this study were deposited in the Sequence Read Archive (SRA) at NCBI under

## Abstract

Microorganisms living in and on macroorganisms may produce microbial volatile compounds (mVOCs) that characterise organismal odours. The mVOCs might thereby provide a reliable cue to carnivorous enemies in locating their host or prey. Parasitism by parasitoid wasps might alter the microbiome of their caterpillar host, affecting organismal odours and interactions with insects of higher trophic levels such as hyperparasitoids. Hyperparasitoids parasitise larvae or pupae of parasitoids, which are often concealed or inconspicuous. Odours of parasitised caterpillars aid them to locate their host, but the origin of these odours and its relationship to the caterpillar microbiome are unknown. Here, we analysed the odours and microbiome of the large cabbage white caterpillar *Pieris brassicae* in relation to parasitism by its endoparasitoid *Cotesia glomerata*. We identified how bacterial presence in and on the caterpillars is correlated with caterpillar odours and tested the attractiveness of parasitised and unparasitised caterpillars to the hyperparasitoid *Baryscapus galactopus*. We manipulated the presence of the external microbiome and the transient internal microbiome of caterpillars to identify the microbial origin of odours. We found that parasitism by *C. glomerata* led to the production of five characteristic volatile products and significantly affected the internal and external microbiome of the caterpillar, which were both found to have a significant correlation with caterpillar odours. The preference of the hyperparasitoid was correlated with the presence of the external microbiome. Likely, the changes in external microbiome and body odour after parasitism were driven by the resident internal microbiome of caterpillars, where the bacterium *Wolbachia* sp. was only present after parasitism. Micro-injection of *Wolbachia* in unparasitised caterpillars increased hyperparasitoid attraction to the caterpillars compared to untreated caterpillars, while no differences were found compared to parasitised caterpillars. In conclusion, our results indicate that host-parasite

Bioproject PRJNA878850. zOTU count data per sample for the internal and external microbiome of caterpillars (10.6084/m9.figshare.21253455), the corresponding measured VOCs (10.6084/m9.figshare.21253518) and the outcome of choice tests (10.6084/m9.figshare.21253524) and no-choice tests (10.6084/m9.figshare.22133318) are available on Figshare.

**Funding:** This work was supported by NWO (ALW Open Programme ALWOP.343 to M.D., Spinoza award SPI 87-313 to M.D., and ALW Open Programme ALWOP.368 to E.H.P) and the Flemish Fund for Scientific Research (FWO; G.0961.19N to B.L. and H.J.). The funders had no role in study design, data collection and analysis, decision to publish, or preparation of the manuscript.

**Competing interests:** The authors have declared that no competing interests exist.

interactions can affect multi-trophic interactions and hyperparasitoid olfaction through alterations of the microbiome.

## Author summary

Bacteria living in and on macroorganisms produce the majority of their odours and thereby play an essential role in species interactions. For example, the hyperparasitoid enemies of parasitic wasps can reliably use odours of parasitised caterpillars to find the parasitic wasp larvae developing inside the caterpillar. We studied whether parasitism of caterpillars caused changes in caterpillar odours and associated internal and external bacterial communities to elucidate whether specific bacteria are responsible for the odours emitted by parasitised caterpillars. We identified which bacteria corresponded with odour production and whether hyperparasitoids could still find parasitised caterpillars that had their bacterial communities experimentally disrupted. We found that parasitised caterpillars have different odour profiles compared to unparasitised caterpillars and that bacteria in and on caterpillars correspond with these changes. The hyperparasitoid was less attracted to parasitised caterpillars with a disrupted skin microbiome. A characteristic bacterium, *Wolbachia* sp., which was only found inside parasitised caterpillars, may have initiated changes in the external microbiome and effects on odour emission. Furthermore, we show that micro-injection of *Wolbachia* in unparasitized caterpillars enhanced hyperparasitoid attraction. Hence, our results indicate that parasitoid wasps indirectly reveal themselves to their hyperparasitoid natural enemies by alteration of the bacterial communities of their host.

## Introduction

It is widely accepted that higher organisms like plants and animals harbour complex microbial communities, including (but not limited to) bacteria, viruses and fungi [1,2]. Organisms are no longer viewed individually, but as "holobionts", referring to the sum of an organism and its full community of microbial associates [3–5]. The microorganismal part of the holobiont (also called microbiome) is known to play a role in the extended phenotype of macroorganisms and can affect host physiology, phenotype and fitness [2,5,6]. These interactions between microorganisms and their host range from mutualistic to pathogenic and parasitic [7].

Insects are among the most speciose and diverse groups of organisms on earth. They overcame many constraints in adaptation to new niches by engaging in mutualistic interactions with microorganisms [7,8]. Besides affecting the host, the insect microbiome can play an important role in intra-and inter-species interactions, extending to interactions between kingdoms [9]. Microorganisms can affect insect olfactory guided behaviour through the production of microorganism-derived volatile organic compounds (mVOCs) [10]. Many insects use mVOCs to navigate a space and find food sources [11], suitable mating partners [12], prey or hosts [13,14].

A substantial amount of animal-associated odours like pheromones, faeces and body odours are tightly associated with the production of mVOCs [10,15]. Organismal odours can contain precise information due to the strong correlation between organismal characteristics and the microbiome, including information about animal identity, sex and life stage [16–18]. Hence, carnivorous and hematophagous insects can eavesdrop upon mVOCs to locate their

prey or (blood) host [13,16,17]. Their preference is often linked to the presence of specific microorganisms producing mVOCs [14,19,20]. For example, mosquitoes are known to be attracted towards individuals with a higher bacterial abundance and lower diversity [13,21]. Bacteria-derived mVOCs are also known to contribute to the characteristic odours of insect faeces (also referred to as frass) [22], which can be used by natural enemies to locate their host or prey [23–25].

Pathogenic microorganisms that challenge the immune system can affect the host microbiome, resulting in fitness consequences when interfering in essential symbioses [26–28]. Microbiome changes can subsequently affect animal-associated odours through an altered production of mVOCs [15]. Organisms with a parasitic lifestyle are known to harness (pathogenic) microorganisms to overcome host immunity [28,29]. A prominent group of insects with a parasitic lifestyle are parasitoid wasps (also called parasitoids) that lay their eggs in or on the body of their host, which are often larval stages of other insects [30]. Koinobiont parasitoids have evolved sophisticated host manipulations due to their intimate relationship with their host. They are highly specialized and let their hosts continue to feed and grow for a significant time after parasitisation [31]. Upon parasitoid oviposition, microbial symbionts such as bacteria and viruses are injected to manipulate host physiology and immune system to benefit the survival of the parasitoid's offspring [32,33]. This is particularly known for caterpillars where parasitism affects the gut microbiome [34,35], which otherwise consists for a large part of plant- and soil-derived transient bacteria [36,37].

Caterpillars of the small cabbage white *Pieris rapae* (Lepidoptera: Pieridae) are known to have altered body odours after parasitism by the koinobiont endoparasitoid wasp *Cotesia glomerata* (Hymenoptera: Braconidae) [38]. This allows the hyperparasitoid *Baryscapus galactopus* (Hymenoptera: Eulophidae), a natural enemy of *C. glomerata*, to locate parasitised caterpillars and complete its lifecycle by oviposition in the parasitoid larvae [39,40]. Likewise, the internal microbiome of the closely related large cabbage white *Pieris brassicae* (Lepidoptera: Pieridae) is strongly modified after parasitism, especially its bacterial community composition [41]. It can be hypothesized that the attractiveness of parasitised caterpillars to hyperparasitoids derives from changes in caterpillar odours through the production of mVOCs, as suggested by its altered microbiome. To date, experiments on attractiveness of odours of parasitised caterpillars to hyperparasitoids could not pinpoint whether the odours are derived from the caterpillar body or products such as frass, and what role external or internal microbiomes play in the odour profile of parasitised caterpillars [38,40]. To investigate the origin of parasitoid-induced attractiveness to hyperparasitoids we studied the effects of parasitism on the organismal odours and the bacterial microbiome of *P. brassicae*, while manipulating the presence of the external microbiome and the transient internal microbiome (including frass).

We report that the organismal odours and bacterial communities of *P. brassicae* are significantly affected by parasitism of *C. glomerata*. We identified five signature volatile organic compounds (VOCs) related to parasitised caterpillars. Linking caterpillar odours and bacterial communities allowed us to identify nine zero-radius operational taxonomic units (zOTUs) correlated with the changes in odour profiles. One of these zOTUs, corresponding to *Wolbachia* sp., was exclusively found inside parasitised caterpillars. Furthermore, we found that the hyperparasitoid *B. galactopus* preferred the odours of parasitised caterpillars with an intact external microbiome over caterpillars with a disrupted external microbiome. Moreover, injection of *Wolbachia* sp. into unparasitised caterpillars resulted in enhanced attraction of hyperparasitoids, shown by a shorter time to first contact with the *Wolbachia*-injected caterpillars compared to mock-injected caterpillars. These results indicate that parasitoid wasps reveal themselves indirectly to their hyperparasitoid natural enemies by modifying the bacterial communities of their host.

## Results

### Parasitism status and external microbiome disruption significantly impact caterpillar odours

To identify how parasitism by the endoparasitoid *C. glomerata* affects the odours emitted by *P. brassicae* caterpillars, we sampled fifth instar stage (L5) unparasitised (Pb) and parasitised (Cg) *P. brassicae* caterpillars. We characterised how caterpillar odours were altered by parasitism, and investigated the impact of internal and external odour sources. For both unparasitised and parasitised caterpillars three different treatments were analysed. (i) untreated caterpillars, (ii) starved caterpillars (ST), and (iii) starved then washed caterpillars to disrupt the external microbiome (ST+EMD). We included starvation to remove transient internal bacteria and frass, and starvation followed by disruption of the external microbiome (referred to as external microbiome disruption treatment) to additionally modify the absolute and relative abundance of some microorganisms in the external microbiome of caterpillars. Caterpillars with the same treatment and parasitism status were divided into groups of eight individuals and subjected to a dynamic headspace collection. Directly after the headspace collection, the same caterpillars were subjected to the collection of their external and internal microbiomes (Fig 1).

A total of 33 different volatile organic compounds (VOCs) were identified in caterpillar odours, categorised in six different chemical classes (Table 1). This dataset was subjected to orthogonal partial least squares discriminant analysis (OPLS-DA), which yielded a significant model (CV-ANOVA: $p < 0.001$, $R^2 = 0.344$, $Q^2 = 0.276$), with two predicting and one orthogonal components accounting for 33.70%, 6.81% and 20.50% of the total variance, respectively. Fourteen volatile compounds (Fig 2A, 2B and 2C and Table 1) strongly contributed to differences among treatments according to the OPLS-DA as indicated by VIP (Variable in Projection) scores larger than 1. These were mainly categorised as alcohols (6 compounds) and

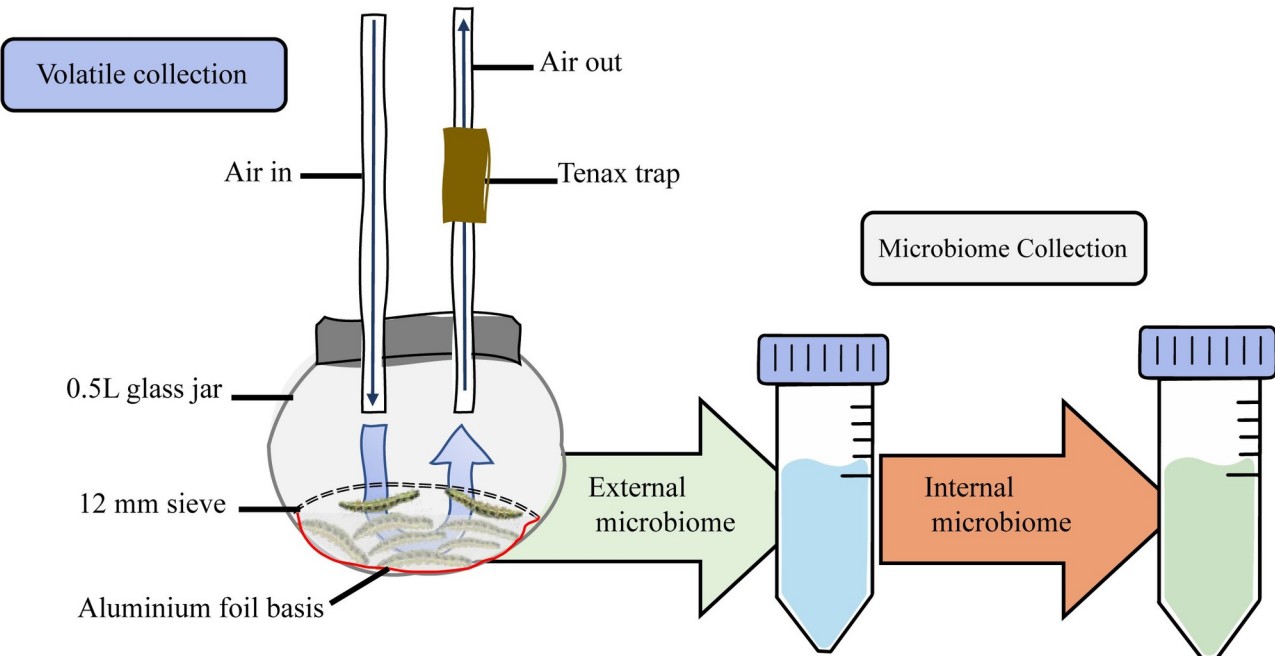

**Fig 1. Experimental setup for the caterpillar odour and microbiome collection.** Groups of eight *C. glomerata* parasitised or unparasitised caterpillars were placed in a glass jar with a restriction device. A dynamic air current was led through the setup and VOCs were trapped in a Tenax trap. After headspace (odour) trapping, the same caterpillars were subjected to the collection of their external and internal microbiome in PBS-Tween80 solutions.

**Table 1. Volatile compounds (tentative identification) detected in the headspace of untreated, starved or starved then external microbiome disrupted unparasitised or *C. glomerata* parasitised caterpillars (*P. brassicae*).** Only those compounds have been included that were present in all replicates of at least one treatment. Amounts of individual compounds are given as the average of peak height /10$^4$ (SE). Variable Importance in the Projection (VIP) values for the OPLS-DA are also given. Bold face VIP scores are higher than 1 and are considered as the most influental VOCs for separation of the treatments as shown in Fig 3. Statistical differences among treatments for compounds with VIP score > 1 are indicated with superscript letters based on Kruskal-Wallis tests with Dunn's test for multiple comparisons including a Bonferroni correction. Compounds in green are correlated with the presence/absence of frass and compounds highlighted in yellow are correlated with parasitism status, compound in blue is not correlated to either of these. Abbreviations used: Cg = Untreated parasitised caterpillars; Cg-ST = starved parasitised caterpillars; Cg-ST+EMD = starved then external microbiome disrupted parasitised caterpillars. Pb = Untreated unparasitised caterpillars; Pb-ST = starved unparasitised caterpillars; Pb-ST+EMD = starved then external microbiome disrupted unparasitised caterpillars.

| Number | Compound and class | RT (min)[1] | ERI | LRI | Pb (n = 10) | Cg (n = 10) | Pb-ST (n = 9) | Cg-ST (n = 10) | Pb-ST+EMD (n = 10) | Cg-ST+EMD (n = 10) | VIP-SCORE |
|---|---|---|---|---|---|---|---|---|---|---|---|
| | **ALCOHOLS** | | | | | | | | | | |
| 1 | 1-Methoxy-2-propanol | 5.8 | 647 | 649 | 1471.9 (411.9) | 833.4 (262.7) | 149.0 (84.6) | 196.6 (46.8) | 674.0 (240.0) | 958.9 (333.0) | 0.69 |
| 2 | 3-Pentanol | 6.5 | 672 | 673[2] | 549.4 (179.6)$^a$ | 261.0 (94.1)$^{ab}$ | 3.2 (2.1)$^c$ | 19.3 (4.0)$^{bc}$ | 3.0 (1.7)$^c$ | 14.7 (5.9)$^{bc}$ | **1.07** |
| 3 | 3-Methyl-3-buten-1-ol | 7.3 | 700 | 710 | 26.4 (6.0)$^a$ | 37.5 (9.4)$^a$ | 2.0 (1.1)$^c$ | 17.2 (3.4)$^{ab}$ | 1.6 (0.8)$^c$ | 6.9 (2.7)$^{bc}$ | **1.26** |
| 4 | 3-Methyl-1-butanol | 7.4 | 705 | 706[4] | 55.6 (10.4)$^a$ | 45.8 (11.5)$^{ab}$ | 8,2 (3.3)$^{bc}$ | 26.8 (7.8)$^{abc}$ | 4.1 (2.4)$^c$ | 51.8 (34.4)$^{abc}$ | **1.02** |
| 5 | (Z)-3-Hexen-1-ol | 10.9 | 856 | 856[4] | 725.0 (173.9)$^a$ | 350.0 (117.9)$^a$ | 2.6 (1.1)$^b$ | 1.4 (0.7)$^b$ | 0.9 (0.6)$^b$ | 1.0 (0.6)$^b$ | **1.17** |
| 6 | 1-Hexanol | 11.2 | 868 | 871 | 59.8 (9.5) | 28.7 (8.6) | 12,7 (3.1) | 16.0 (4.6) | 7.7 (2.5) | 9.8 (3.1) | 0.73 |
| 7 | (Z)-3-Hepten-1-ol | 13.1 | 929 | 947 | 0.1 (0.0)$^c$ | 29.5 (9.0)$^a$ | 0.9 (0.8)$^{bc}$ | 8.8 (3.0)$^{ab}$ | 0.1 (0.0)$^c$ | 4.6 (2.4)$^{bc}$ | **1.45** |
| 8 | 3-Octanol | 15.1 | 997 | 991 | 19.6 (3.0)$^a$ | 34.6 (9.9)$^a$ | 3.4 (1.6)$^b$ | 14.9 (3.9)$^a$ | 1.3 (0.5)$^b$ | 8.5 (3.3)$^{ab}$ | **1.23** |
| | **ALDEHYDES** | | | | | | | | | | |
| 9 | (E)-2-Butenal | 5.4 | 632 | 632 | 46.5 (9.9) | 27.9 (8.3) | 20.2 (4.7) | 14.8 (4.0) | 10.3 (1.8) | 10.8 (1.7) | 0.71 |
| 10 | 3-Methylbutanal | 5.5 | 635 | 637 | 40.1 (7.0) | 20.2 (8.4) | 5.6 (1.4) | 5.0 (1.1) | 3.8 (0.9) | 7.2 (3.8) | 0.76 |
| 11 | 2-Methylbutanal | 5.7 | 643 | 641 | 53.9 (9.9) | 30.8 (13.7) | 6.4 (2.0) | 4.5 (0.9) | 4.8 (1.0) | 6.0 (3.5) | 0.82 |
| 12 | (Z)-2-Pentenal | 8.0 | 736 | 727[5] | 17.4 (3.0)$^a$ | 9.1 (2.6)$^a$ | 0.1 (0.0)$^b$ | 0.3 (0.2)$^b$ | 0.1 (0.0)$^b$ | 0.4 (0.3)$^b$ | **1.18** |
| | **AROMATICS** | | | | | | | | | | |
| 13 | Phenylacetaldehyde | 16.6 | 1053 | 1053 | 22.3 (3.0) | 15.1 (3.6) | 6.7 (1.1) | 7.1 (0.8) | 7.2 (1.3) | 7.7 (1.2) | 0.73 |
| 14 | 1-Phenylethanol | 17.0 | 1067 | 1063 | 10.4 (2.0)$^a$ | 8.0 (2.7)$^a$ | 0.1 (0.0)$^b$ | 0.1 (0.0)$^b$ | 0.1 (0.0)$^b$ | 0.2 (0.1)$^b$ | **1.19** |
| 15 | 2-Phenylethanol | 18.6 | 1122 | 1120 | 43.2 (11.4) | 29.6 (8.5) | 2.5 (1.0) | 3.9 (1.1) | 2.4 (0.8) | 4.1 (1.4) | 0.95 |
| 16 | Benzothiazole | 21.8 | 1249 | 1238 | 174.6 (26.5) | 111.1 (27.5) | 59.7 (17.5) | 71.1 (26.1) | 66.4 (9.3) | 68.5 (15.7) | 0.71 |
| 17 | (E)-Anethole | 23.1 | 1298 | 1294 | 32.0 (6.7) | 19.5 (4.7) | 33.3 (10.7) | 16.5 (5.3) | 18.1 (4.7) | 22.3 (2.7) | 0.48 |
| | **KETONES** | | | | | | | | | | |
| 18 | 2,3-Butanedione | 4.2 | <600 | 588[3] | 704.0 (200.5) | 344.0 (87.3) | 69.1 (33.1) | 87.3 (17.0) | 74.5 (33.3) | 32.3 (8.3) | 0.93 |
| 19 | 3-Pentanone | 6.4 | 667 | 669[4] | 75.4 (35.9)$^a$ | 35.7 (11.2)$^a$ | 3.2 (0.9)$^b$ | 9.7 (0.9)$^a$ | 2.1 (0.7)$^b$ | 10.2 (2.1)$^a$ | **1.10** |
| 20 | 3-Hydroxy-2-butanone | 6.7 | 678 | 681 | 10.1 (2.4) | 8.1 (4.2) | 0.9 (0.5) | 1.0 (0.5) | 0.1 (0.0) | 0.7 (0.3) | 0.95 |
| 21 | 3-Octanone | 14.8 | 986 | 984 | 0.1 (0.0)$^b$ | 2.9 (0.9)$^a$ | 0.1 (0.0)$^b$ | 3.0 (0.5)$^a$ | 0.1 (0.0)$^b$ | 2.7 (0.8)$^a$ | **1.78** |
| | **NITROGEN AND/OR SULPHUR CONTAINING COMPOUNDS** | | | | | | | | | | |
| 22 | 2-Methyl-propanenitrile | 4.8 | 611 | 623 | 14.9 (5.7) | 17.6 (8.5) | 16.4 (9,2) | 14.7 (7.2) | 34.6 (20.4) | 8.2 (2.7) | 0.51 |
| 23 | Dimethyl disulfide | 7.8 | 725 | 727 | 878.4 (258.1) | 843.5 (313.9) | 61.3 (46.8) | 33.0 (9.3) | 23.7 (10.6) | 20.5 (5.4) | 0.97 |
| 24 | Methylthioacetaldehyde | 8.2 | 748 | 760[3] | 19.2 (5.0)$^a$ | 21.7 (11.0)$^a$ | 0.3 (0.2)$^b$ | 0.1 (0.0)$^b$ | 0.1 (0.0)$^b$ | 0.1 (0.0)$^b$ | **1.17** |
| 25 | Dimethyl trisulfide | 14.6 | 982 | 988[3] | 356.8 (123.1)$^a$ | 788.9 (564.3)$^a$ | 5.4 (2.7)$^b$ | 1.7 (0.7)$^b$ | 7.0 (5.6)$^b$ | 3.4 (1.0)$^b$ | **1.08** |
| 26 | 4-(methylthio)-Butanenitrile | 17.6 | 1089 | 1092[3] | 431.6 (209.9)$^a$ | 127.7 (58.3)$^a$ | 8.9 (7.5)$^b$ | 1.3 (1.2)$^b$ | 0.1 (0.0)$^b$ | 2.4 (1.6)$^b$ | **1.14** |

(*Continued*)

**Table 1.** (Continued)

| Number | Compound and class | RT (min)[1] | ERI | LRI | Pb (n = 10) | Cg (n = 10) | Pb-ST (n = 9) | Cg-ST (n = 10) | Pb-ST+EMD (n = 10) | Cg-ST+EMD (n = 10) | VIP-SCORE |
|---|---|---|---|---|---|---|---|---|---|---|---|
| 27 | Methyl (methylthio) methyl disulfide | 19.1 | 1144 | 1144 | 108.9 (39.2)[a] | 194.1 (118.2)[a] | 1.3 (0.5)[b] | 0.3 (0.2)[b] | 0.2 (0.1)[b] | 0.1 (0.0)[b] | **1.21** |
| | **TERPENOIDS** | | | | | | | | | | |
| 28 | p-Menth-3-ene | 14.9 | 992 | 988 | 14.2 (3.6) | 11.8 (5.5) | 10.1 (4.7) | 12.1 (4.7) | 14.6 (4.2) | 13.8 (6.1) | 0.49 |
| 29 | 1,8-Cineole | 16.4 | 1044 | 1042 | 18.9 (3.2) | 11.3 (2.6) | 1.8 (1.1) | 2.8 (0.4) | 1.7 (0.7) | 2.8 (0.6) | 0.99 |
| 30 | Dihydromyrcenol | 17.2 | 1073 | 1074 | 26.4 (6.1) | 14.4 (3.3) | 6.4 (1.6) | 7.7 (1.3) | 5.0 (0.7) | 7.5 (1.7) | 0.89 |
| 31 | Isomenthol | 20.3 | 1189 | 1192 | 17.9 (1.9) | 12.3 (2.5) | 8.3 (2.4) | 8.3 (1.8) | 5.8 (1.1) | 7.3 (0.9) | 0.73 |
| 32 | Verbenone | 21.2 | 1225 | 1223 | 4.2 (0.5) | 3.2 (0.6) | 2.1 (0.6) | 2.4 (0.5) | 1.4 (0.5) | 1.9 (0.5) | 0.80 |
| 33 | Bornyl acetate | 23.1 | 1302 | 1304 | 15.6 (8.9) | 7.6 (2.5) | 5.7 (1.2) | 10.5 (2.8) | 6.0 (1.6) | 12.1 (4.2) | 0.82 |

ERI: Experimentally obtained retention indices on a ZB-5MS analytical column

LRI: Retention indices obtained from NIST MS library (https://webbook.nist.gov/), on a column with (5%-Phenyl)-methylpolysiloxane stationary phase or equivalent unless stated otherwise.

[1]RT: Retention time (in minutes) of compounds in the chromatographic window.

[2]LRI: Retention indices [42]

[3]LRI: Retention indices [38]

[4]LRI: Retention indices [43]

[5]LRI on a 100% polydimethylsiloxane (PDMS) or equivalent stationary phase.

nitrogen and/or sulphur containing compounds (4 compounds). Parasitism status significantly affected caterpillar odours (PERMANOVA: $F_{1,53}$ = 4.594, $p$ = 0.011), from which the top four VOCs in VIP-score (3-octanone, ($Z$)-3-hepten-1-ol, 3-methyl-3-buten-1-ol and 3-octanol) and 3-pentanone were characteristic for parasitised caterpillars (Table 1 highlighted in yellow and Fig 2A,). A compound was considered characteristic for parasitised caterpillars when it was significantly increased for parasitised caterpillars in at least one pairwise comparison with unparasitised caterpillars undergoing the same treatment (e.g. Cg-ST and Pb-ST).

The included treatments allowed us to address the origin of these compounds. Starvation (ST) and starvation followed by external microbiome disruption (ST+EMD) significantly affected caterpillar body odours (PERMANOVA: $F_{2,53}$ = 17.292, $p$ < 0.001), but differences between parasitised and unparasitised caterpillars were still present. Pairwise analyses within starved treatments (Pb-ST vs Cg-ST) and starved then external microbiome disrupted caterpillars with different parasitism statuses (Pb-ST+EMD vs Cg-ST+EMD) highlight the importance of 3-octanone, ($Z$)-3-hepten-1-ol, 3-methyl-3-buten-1-ol, 3-octanol and 3-pentanone in characterising parasitism status of caterpillars (S1 and S2 Figs and Table A in S1 Supporting Information). This indicates that the source of these VOCs was elsewhere than the frass and skin of the caterpillar, and that they are likely formed internally in the caterpillar.

Overall, starvation followed by external microbiome disruption (ST+EMD) did not impact caterpillar odours strongly, compared to only starvation (ST). Only 3-methyl-3-buten-1-ol was found to be significantly reduced when comparing starved and starved then external microbiome disrupted treatments of the same parasitism status. Accordingly, the separation in OPLS-DA was less evident between starved and starved then external microbiome disrupted caterpillars with the same parasitism status (Figs 3A, S3 and S4). Untreated caterpillars produced several compounds at a higher level (Figs 2B and 3B), including most nitrogen- and/or sulphur-containing compounds. This is likely due to the presence of frass during the headspace collection of untreated caterpillars. Whether hyperparasitoid attraction to caterpillar odours could be explained by frass was investigated in a two-chamber olfactometer setup as

 Parasitism changes the odours and microbiome of caterpillars which reveals them to hyperparasitoids

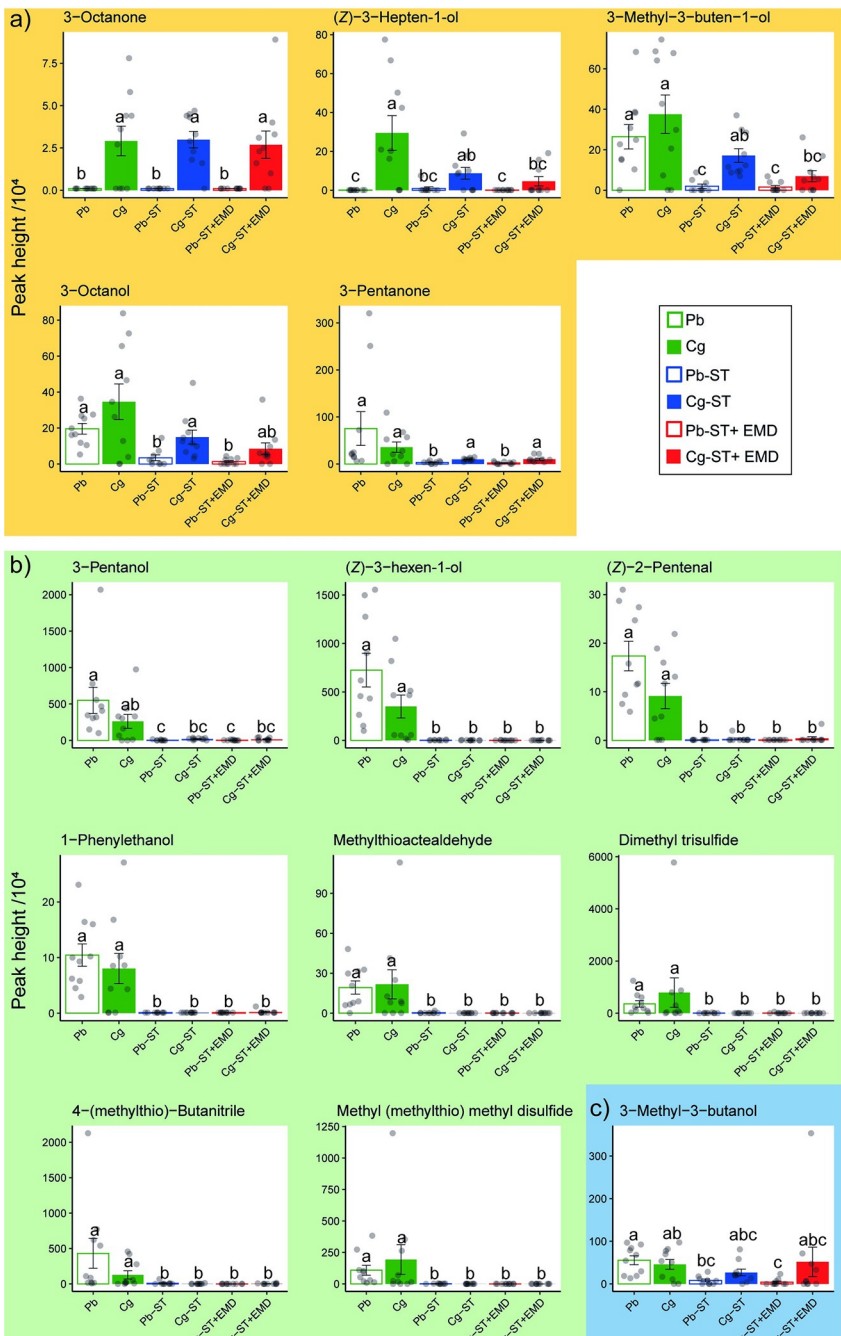

**Fig 2. Volatile compounds (tentative identification) detected in the headspace of untreated, starved or starved then external microbiome disrupted unparasitised or *C. glomerata* parasitised caterpillars (*P. brassicae*) with a Variable Importance in the Projection (VIP) values > 1 for the OPLS-DA.** Amounts of individual compounds are given as the average of peak height /10$^4$ (±SE). Statistical differences among treatments for compounds with VIP score > 1 are indicated with different letters based on Kruskal-Wallis tests with Dunn's test for multiple comparisons including a Bonferroni correction. a) Compounds in yellow background are correlated with parasitism status. b) Compounds in green background are correlated with the presence/absence of frass. c) Compound in blue background is not correlated to either of these. Abbreviations used: Pb = Untreated unparasitised caterpillars; Cg = Untreated parasitised caterpillars; Pb-ST = starved unparasitised caterpillars; Cg-ST = starved parasitised caterpillars; Pb-ST +EMD = starved then external microbiome disrupted unparasitised caterpillars. Cg-ST+EMD = starved then external microbiome disrupted parasitised caterpillars.

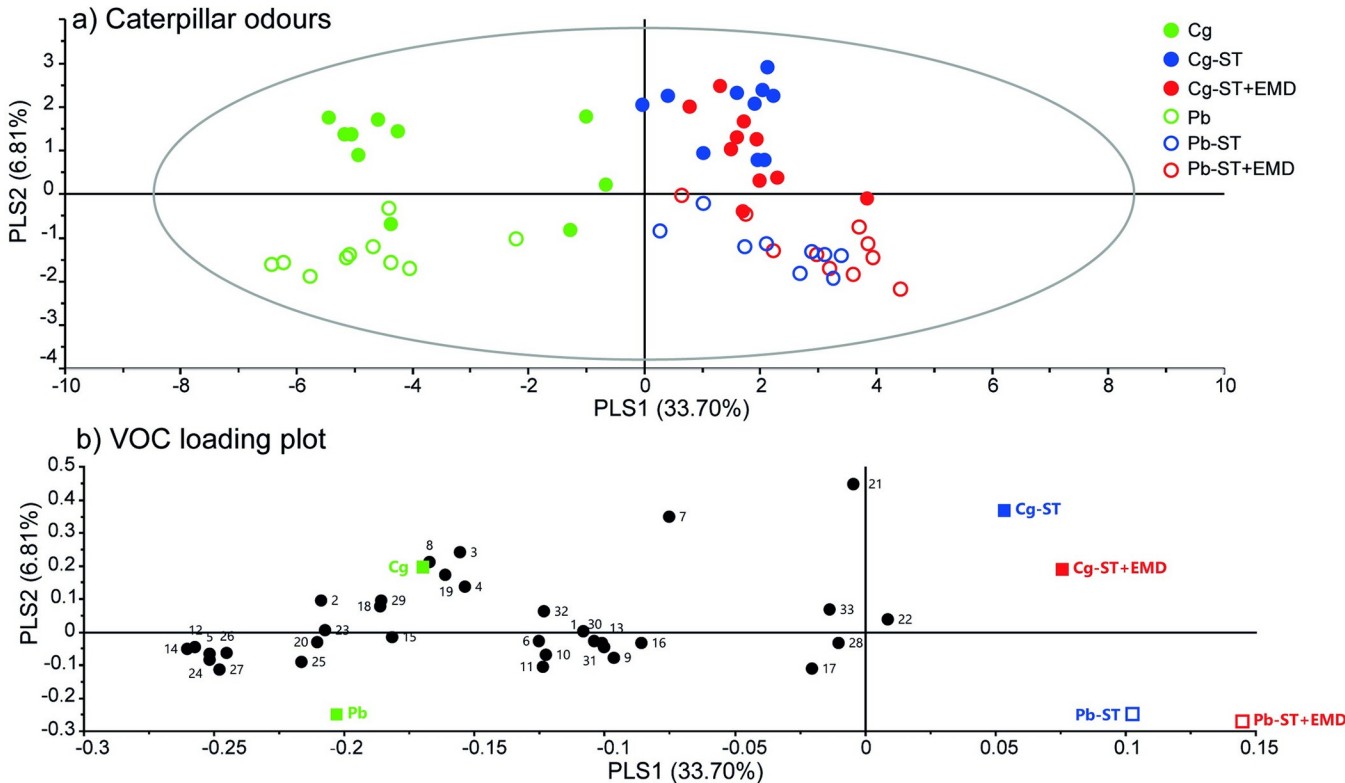

**Fig 3. Overview of caterpillar volatile profiles of untreated, starved or starved then external microbiome disrupted *C. glomerata* parasitised or unparasitised caterpillars (*P. brassicae*).** a) OPLS-DA (Orthogonal Projection to Latent Structures Discriminant Analysis) plot for the volatile blends of different groups of caterpillars. The Hotelling's T2 ellipse confines the confidence region (95%) of the score plot. b) Loading plot defining the contribution of each of the volatile compounds to the separation of treatment groups. Volatile compounds closer to a treatment in the plot correlate stronger with the treatment. For compound identity see Table 1. Abbreviations used: Cg = Untreated parasitised caterpillars; Cg-ST = starved parasitised caterpillar; Cg-ST +EMD = starved then external microbiome disrupted parasitised caterpillars. Pb = Untreated unparasitised caterpillars; Pb-ST = starved unparasitised caterpillars; Pb-ST+EMD = starved then external microbiome disrupted unparasitised caterpillars.

described in [38]. Our results indicate that frass was attractive over an empty chamber (Empty vs Pb-frass: $p = 0.007$, U = 643.5; Empty vs Cg-frass: $p = 0.163$, U = 565), but reject the hypothesis that the hyperparasitoid can use frass to discriminate between parasitised and unparasitised caterpillars (Pb-frass vs Cg-frass: $p = 0.504$, U = 1066.5) (S5 Fig).

## Parasitism status and external microbiome disruption significantly impact caterpillar external and internal microbiome

Because organismal odours are tightly associated with the microorganisms present in or on the body of macroorganisms, we subsequently characterised how the caterpillar microbiome was altered by parasitism, starvation, and starvation followed by external microbiome disruption. Directly after the collection of caterpillar odours, the same caterpillars were subjected to the collection of their external and internal microbiomes (Fig 1). Analysis of the bacterial 16S ribosomal RNA (rRNA) gene sequences revealed a significant effect of parasitism on the external microbiome (PERMANOVA, $F_{1,53} = 5.759$, $p = 0.002$) of *P. brassicae* caterpillars (Fig 4A). The external microbiome of all caterpillars mainly consisted of three zOTUs (combined >69% relative abundance); *Enterococcus* sp. (zOTU 1), Enterobacteriaceae (zOTU 2) and *Staphylococcus* sp. (zOTU 3) (Fig 4C and Table B in S1 Supporting Information). *Staphylococcus* sp. (zOTU 3) was more abundant on unparasitised caterpillars, while on parasitised caterpillars an

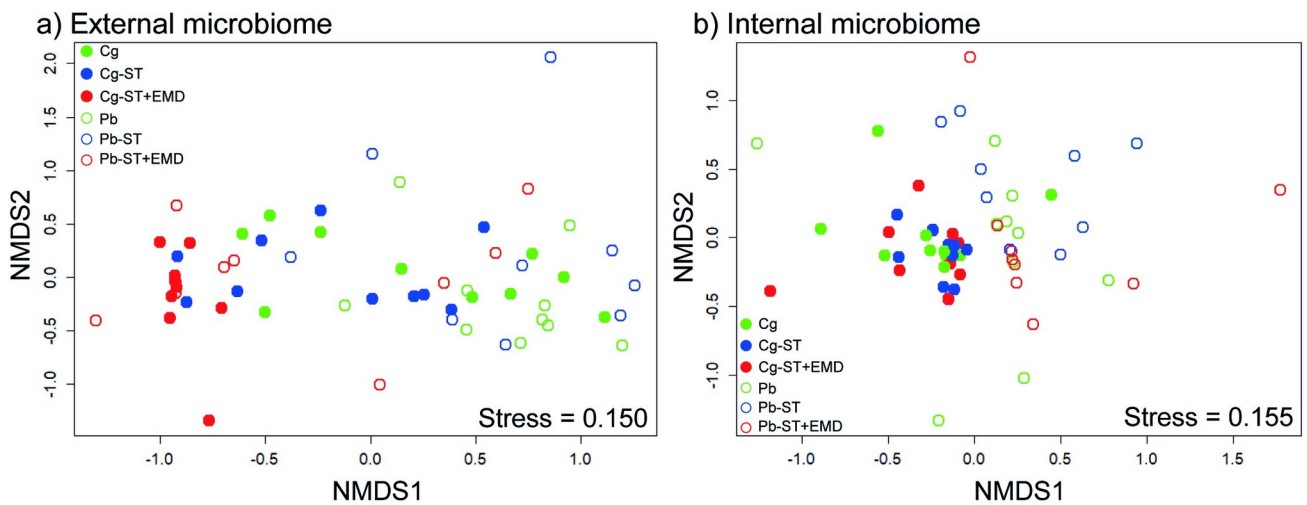

## c) External microbiome

| Blasted ID | zOTU ID | Pb | Cg | Pb-ST | Cg-ST | Pb-ST+EMD | Cg-ST+EMD | Overall |
|---|---|---|---|---|---|---|---|---|
| *Enterococcus* | zOTU1 | 32.5 | 39.7 | 23.1 | 48.2 | 58.9 | 72.4 | 45.8 |
| Enterobacteriaceae | zOTU2 | 7.4 | 27.7 | 10.3 | 32.9 | 25.5 | 26.5 | 21.7 |
| *Staphylococcus* | zOTU3 | 29.1 | 10.0 | 27.8 | 9.9 | 3.5 | 0.0 | 13.4 |
| *Jeotgalicoccus* | zOTU7 | 6.3 | 4.0 | 4.0 | 2.8 | 0.5 | 0.0 | 2.9 |
| *Acinetobacter* | zOTU5 | 1.6 | 4.3 | 7.6 | 1.5 | 0.9 | 0.0 | 2.7 |
| *Staphylococcus* | zOTU11 | 7.4 | 1.4 | 5.2 | 0.4 | 1.3 | 0.0 | 2.6 |
| *Corynebacterium* | zOTU10 | 4.5 | 3.6 | 3.4 | 1.9 | 0.2 | 0.0 | 2.3 |
| *Acinetobacter* | zOTU6 | 2.6 | 2.2 | 4.0 | 0.1 | 1.2 | 0.0 | 1.7 |
| *Glutamicibacter* | zOTU12 | 3.0 | 2.6 | 2.3 | 1.4 | 1.3 | 0.0 | 1.7 |

## d) Internal microbiome

Prevalence (%) 100 80 60 40 20 0

| Blasted ID | zOTU ID | Pb | Cg | Pb-ST | Cg-ST | Pb-ST+EMD | Cg-ST+EMD | Overall |
|---|---|---|---|---|---|---|---|---|
| *Enterococcus* | zOTU1 | 85.1 | 79.9 | 72.5 | 84.3 | 86.9 | 85.8 | 82.4 |
| Enterobacteriaceae | zOTU2 | 11.5 | 11.0 | 17.6 | 8.9 | 9.7 | 7.6 | 11.1 |
| *Wolbachia** | zOTU4 | 0.0 | 8.2 | 0.0 | 6.5 | 0.0 | 6.3 | 3.5 |
| *Sphingomonas* | zOTU9 | 0.2 | 0.3 | 8.9 | 0.1 | 2.1 | 0.0 | 2.0 |

**Fig 4. Overview of caterpillar-associated bacterial communities of untreated, starved or starved then external microbiome disrupted *C. glomerata* parasitised or unparasitised caterpillars (*P. brassicae*).** a) Non-metric multidimensional scaling (NMDS) ordination plots based on Bray–Curtis distances of Hellinger-transformed relative abundance data of the external bacterial communities. b) NMDS ordination for the internal bacterial communities. c) External bacterial community profiles of the different caterpillar samples. d) Internal bacterial community profiles of the different caterpillars samples. For each zOTU, the average relative abundance (%) for each group is given in the cell as a percentage, whereas the colour indicates prevalence (white is absent). Only bacteria with an overall relative abundance >1.5% are shown in (c) and (d). The full overview is provided as supporting information (Table B in S1 Supporting Information). zOTUs are identified by a BLAST search against type materials in GenBank. When no significant similarity was found with type materials, the BLAST analysis was performed against entire GenBank (indicated with an asterisk). Identifications were performed at genus level; when identical scores were obtained for different genera, identifications were performed at family level. Abbreviations used: Cg = untreated parasitised caterpillars; Cg-ST = starved parasitised caterpillar; Cg-ST+EMD = starved then external microbiome disrupted parasitised caterpillars. Pb = untreated unparasitised caterpillars; Pb-ST = starved unparasitised caterpillars; Pb-ST+EMD = starved then external microbiome disrupted unparasitised caterpillars.

unidentified member of Enterobacteriaceae (zOTU 2) had a higher relative abundance and the relative abundance of *Staphylococcus* sp. (zOTU 3) was decreased. Moreover, parasitism had a significant effect on the internal (PERMANOVA, $F_{1,53} = 4.865$, $p = 0.014$) bacterial community supporting a previous study [41] (Fig 4B). The internal microbiome was dominated by *Enterococcus* sp. (zOTU 1) and Enterobacteriaceae (zOTU 2) (combined >90% of relative abundance), but *Staphylococcus* sp. (zOTU 3) was absent. *Wolbachia* sp. (zOTU 4) was only found in parasitised caterpillars (Fig 4D and Table B in S1 Supporting Information).

To further elucidate where the differences in microbiome came from, we starved caterpillars to exclude transient and frass-related microorganisms and, starved then external microbiome disrupted caterpillars to affect microorganisms on the skin as well. Treatment had a significant effect on the external bacterial community (PERMANOVA, $F_{2,53} = 5.716$, $p < 0.001$). A strong cluster of bacteria collected from parasitised, starved then external microbiome disrupted caterpillars (Cg-ST+EMD) is clearly visible in the NMDS ordination (Fig 4A). *Staphylococcus* sp. (zOTU 3) almost fully declined after starvation followed by external microbiome disruption and the relative abundance of *Enterococcus* sp. (zOTU 1) increased. The internal bacterial communities were not substantially affected by treatment (PERMANOVA, $F_{2,53} = 1.083$, $p = 0.341$), indicating that the difference in microbiome between unparasitised and parasitised caterpillars is caused by resident bacteria. Simply put, the parasitism signal in the microbiome was captured in the resident internal microbiome, where *Wolbachia* sp. (zOTU 4) was only present in parasitised caterpillars. On the external microbiome there was a remarkable decrease in the relative abundance of *Staphylococcus* sp. (zOTU 3) after parasitism. The Shannon diversity indices of the external and internal microbiomes were significantly affected by parasitism (external: $F_{1,53} = 5.584$, $p = 0.022$; internal: $F_{1,53} = 4.379$, $p = 0.041$; ANOVA). The diversity of the external bacterial community decreased after parasitism and the diversity of the internal bacterial community increased. Treatment of caterpillars reduced the diversity of the external, but not the internal microbiome (external: $F_{2,53} = 12.661$, $P < 0.001$; internal: $F_{2,53} = 2.394$, $p = 0.062$; ANOVA) (S7 Fig).

## Linking microbial communities to caterpillar odours

To study the relationship between caterpillar odours and bacteria, we decided to subset our data by omitting the untreated samples. The presence of frass in these samples was shown to strongly impact the odours, but had a limited effect on the signature odours for parasitism (Table 1), the internal microbiome (Fig 4B) and hyperparasitoid preference in a two-chamber olfactometer setup (S5 Fig). The relationship between the caterpillar microbiome and odours was found to be significant for both the external (ANOVA-like permutation test: $F_{5,33} = 2.704$, $R^2 = 0.29$, $p = 0.001$) and internal bacterial communities (ANOVA-like permutation test: $F_{4,34} = 2.99$, $R^2 = 0.26$, $p = 0.001$). A total of nine zOTUs were selected, which contributed most to the variation in odour composition between samples. These zOTUs represented five zOTUs from the external microbiome and four from the internal microbiome, indicated by the vectors in the RDA tri-plots (Fig 5A and 5B). There was no overlap in the selected zOTUs for the internal- and external bacterial communities.

From the selected bacteria located in the internal microbiome *Wolbachia* sp. (zOTU 4) was the most abundant and was only present in parasitised caterpillars. *Sphingomonas* sp. (zOTU 9) had a higher relative abundance in treated unparasitised caterpillars, and was nearly absent in parasitised caterpillars. A so far unidentified bacterium (zOTU 94) and *Proteus* sp. (zOTU 220) had a low relative abundance (≤0.1%) and thus likely had a negligible effect on the caterpillar odours (Fig 5D). Accordingly, all selected zOTUs for the external microbiome had a low (≤0.2%) overall relative abundance. From these, *Chryseobacterium* sp. (zOTU 49),

          

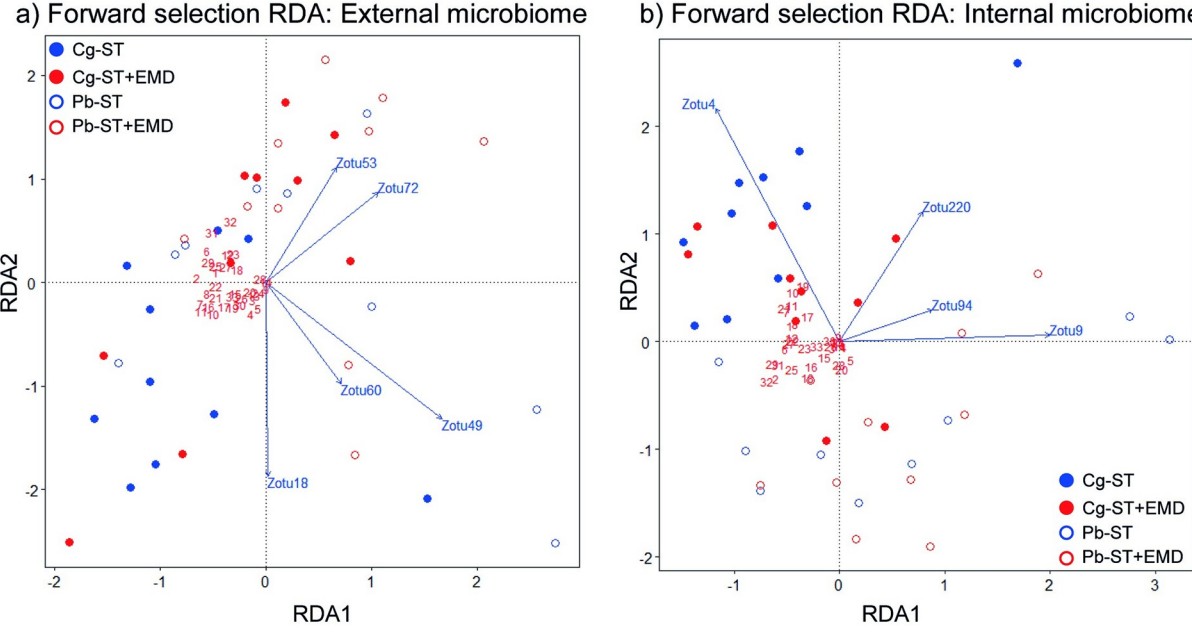

## c) Selected zOTUs external microbiome

| Blasted ID | zOTU ID | Pb-ST | Cg-ST | Pb-ST+EMD | Cg-ST+EMD | Overall |
|---|---|---|---|---|---|---|
| *Bradyrhizobium* | **zOTU18** | 0.3 | 0.2 | 0.0 | 0.4 | 0.2 |
| *Chryseobacterium* | **zOTU49** | 0.2 | 0.0 | 0.0 | 0.0 | 0.0 |
| *Pseudomonas* | **zOTU53** | 0.2 | 0.0 | 0.1 | 0.0 | 0.0 |
| *Enterococcus* [98.7%] | **zOTU60** | 0.0 | 0.0 | 0.0 | 0.0 | 0.0 |
| *Paenalcaligenes* | **zOTU72** | 0.0 | 0.0 | 0.4 | 0.0 | 0.1 |

## d) Selected zOTUs internal microbiome

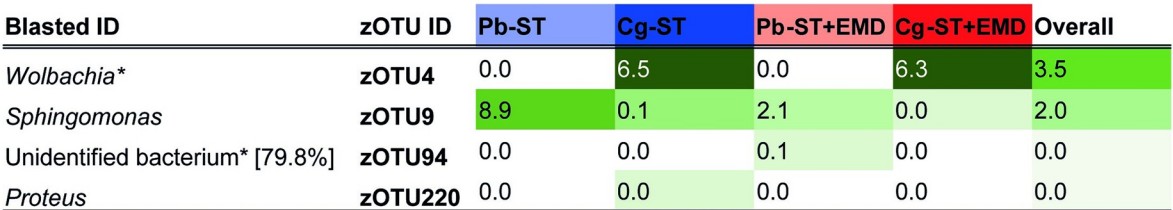

Prevalence %
100 80 60 40 20 0

| Blasted ID | zOTU ID | Pb-ST | Cg-ST | Pb-ST+EMD | Cg-ST+EMD | Overall |
|---|---|---|---|---|---|---|
| *Wolbachia** | **zOTU4** | 0.0 | 6.5 | 0.0 | 6.3 | 3.5 |
| *Sphingomonas* | **zOTU9** | 8.9 | 0.1 | 2.1 | 0.0 | 2.0 |
| Unidentified bacterium* [79.8%] | **zOTU94** | 0.0 | 0.0 | 0.1 | 0.0 | 0.0 |
| *Proteus* | **zOTU220** | 0.0 | 0.0 | 0.0 | 0.0 | 0.0 |

**Fig 5. Forward selection redundancy analyses (RDA) on volatiles of untreated, starved or starved then external microbiome disrupted *C. glomerata* parasitised or unparasitised caterpillars (*P. brassicae*) and their bacterial communities.** The triplots indicate the volatile composition of the caterpillars (dots), the volatile compounds (numbers as in Table 1) and forward selected zOTUs after 9999 permuations (vectors) for the external (a) and internal (b) bacterial communities. Further, relative abundance-prevalence data of the selected zOTUs are presented for the external (c) and internal (d) bacterial community. The average relative (%) abundance for each group is given in the cell as a percentage, whereas the colour indicates prevalence (white is absent). zOTUs were identified by a BLAST search against type materials in GenBank. When no significant similarity was found with type materials, the BLAST analysis was performed against entire GenBank (indicated with an asterisk). Identifications were performed at genus level. When identity percentages were lower than 99%, the percentage of sequence identity with the GenBank entry is given between brackets. Abbreviations used: Cg-ST = starved parasitised caterpillar; Cg-ST+EMD = starved then external microbiome disrupted parasitised caterpillars; Pb-ST = starved unparasitised caterpillars; Pb-ST+EMD = starved then external microbiome disrupted unparasitised caterpillars.

*Enterococcus* sp. (zOTU 53) and *Paenalcaligenes* sp. (zOTU 72) were exclusively found on unparasitised caterpillars, whereas *Bradyrhizobium* sp. (zOTU 18) had an increased relative abundance on parasitised caterpillars (Fig 5C). Interestingly, Enterobacteriaceae sp. (zOTU 2) and *Staphylococcus* sp. (zOTU 3) were highly abundant in the external microbiome and affected by parasitism in their relative abundance, but were not selected in the forward selection RDA (Fig 4C). These results point towards a microbe-derived signal of parasitism that comes from within the caterpillar and determines caterpillar odours, while external bacterial communities and frass are less likely to play a role in the production of volatile compounds associated with parasitism status.

## Hyperparasitoid females are no longer attracted to external microbiome disrupted parasitised caterpillars

A total of 1000 *B. galactopus* females were tested in two-choice assays in a Y-tube olfactometer for their preference to caterpillar odours. Hyperparasitoids did respond to caterpillar odours and 84.4% of the tested females made a choice within 10 min (Fig 6). The hyperparasitoid preferred the odours of starved, parasitised *P. brassicae* (Cg-ST) over an empty arm (binomial test, $p < 0.001$) and over odours from caterpillars that had received the external microbiome disruption treatment after starvation (Cg-ST+EMD) (binomial test, $p < 0.001$) (Fig 6). Disruption of the external microbiome caused a loss of preference for caterpillar odours. Hyperparasitoid preference did not differ between differently treated unparasitised caterpillars (Pb-ST +EMD vs Pb-ST; binomial test, $p = 0.28$) or starved then external microbiome disrupted caterpillars with different parasitism statuses (Pb-ST+EMD vs Cg-ST+EMD; binomial test, $p = 0.06$). When the odours of unparasitised versus parasitised starved caterpillars were offered, no preference was found (Pb-ST vs Cg-ST; binomial test, $p = 0.76$) contrasting the

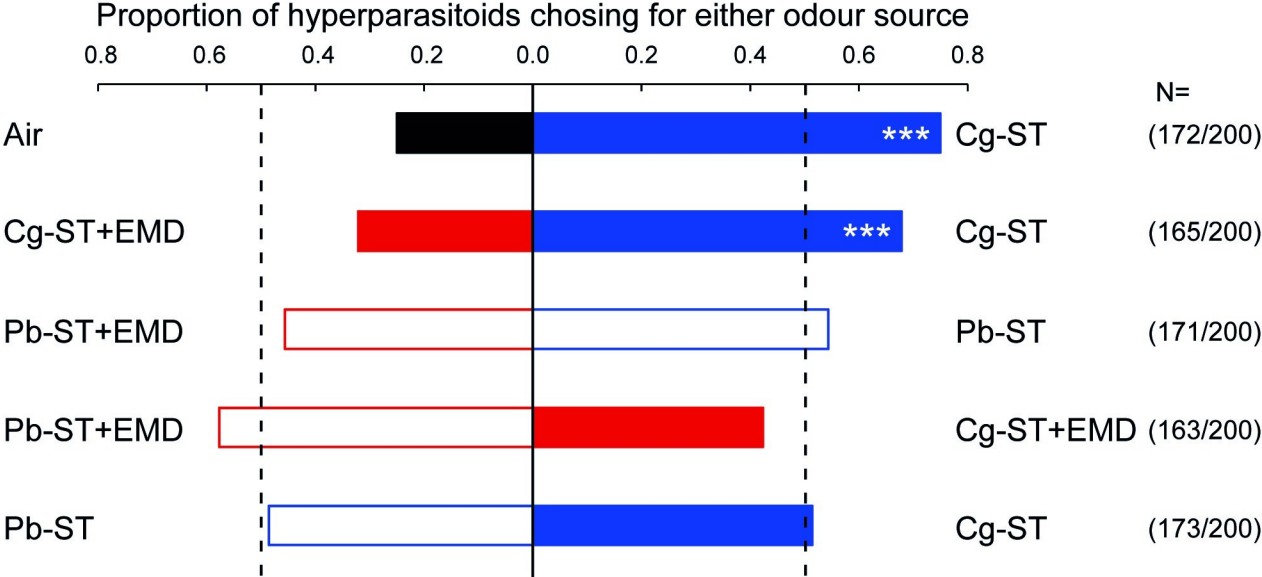

**Fig 6. Preference of the hyperparasitoid *B. galactopus* for caterpillar body odours of *C. glomerata* parasitised and unparasitised *P. brassicae* caterpillars.** This was tested with Y-tube olfactometer tests. Tested combinations were selected according to the hypothesis that hyperparasitoids can use caterpillar body odours, which are (at least partially) determined by the external microbiome. Numbers between brackets indicate the number of wasps that made a choice within 10 min from the start of the experiment versus the total number of wasps tested. *** $P < 0.001$; two-sided binomial test. Abbreviations used: Air = clean air (black); Cg-ST = starved parasitised caterpillar (solid blue); Cg-ST+EMD = starved and external microbiome disrupted parasitised caterpillars (solid red). Pb-ST = starved unparasitised caterpillars (blue border); Pb-ST+EMD = starved then external microbiome disrupted unparasitised caterpillars (red border).

comparisons of their microbiome, odour profiles (S3 and S4 Figs) and earlier findings in a closely related caterpillar species (*P. rapae*) [38]. Highly similar results were found when we repeated this experiment using the same species of caterpillar and hyperparasitoid, but parasitised the caterpillars with *Hyposoter ebeninus,* which is another parasitoid host species of the hyperparasitoid (S6 Fig). The consistency of these results indicate that the preference of the hyperparasitoid is correlated with the presence of the external microbiome, though it is not discriminating between the odours of parasitised and unparasitised caterpillars in this setup.

## Direct evidence for the role of *Wolbachia* in hyperparasitoid host location

In order to assess whether the presence of *Wolbachia* affects hyperparasitoid behaviour, a no-choice experiment was performed in which *B. galactopus* individuals were subjected to *P. brassicae* caterpillars injected with *Wolbachia* sp. (Wolb-PBS, injected once with *Wolbachia*; Wolb-Wolb, injected twice with *Wolbachia*) in comparison with caterpillars parasitised by *C. glomerata* (Parasitised-PBS) or unparasitised caterpillars (Unparasitised-PBS) (Fig 7). Within one hour of release, *B. galactopus* responded by mounting caterpillars in 84.7% of the cases, and the lowest response rate was observed for unparasitised individuals (75.0%). Significant differences in the time used to make first contact and mount the caterpillars were found between treatments (Kruskal-Wallis $\chi^2$ = 11.24, $p$ = 0.010). Pairwise comparisons showed significant differences in time to first contact between unparasitised and parasitised caterpillars (Dunn's test $Z$ = 2.95, $p$ = 0.019) and between unparasitised caterpillars and those injected once with *Wolbachia* ($Z$ = 2.62, $p$ = 0.043). On average, the wasps needed 21 min 02 sec before mounting unparasitised caterpillars, while it took less time, to mount parasitised caterpillars (10 min 27 sec), caterpillars injected once with *Wolbachia* (12 min 16 sec) and caterpillars injected two times with *Wolbachia* (16 min 36 sec), respectively (Fig 7A). Differences between treatments were also statistically significant for the time spent mounting (Kruskal-Wallis $\chi^2$ = 10.241, $p$ = 0.017). Pairwise comparisons revealed statistically significant differences in total mounting time between unparasitised and parasitised caterpillars ($Z$ = -2.85, $p$ = 0.026), and between parasitised caterpillars and those injected once with *Wolbachia* ($Z$ = 2.77, $p$ = 0.027). On average, hyperparasitoids spent 57 min 55 sec on parasitised individuals, while they spent 50 min 50 sec on healthy caterpillars, 50 min 22 sec on caterpillars injected once with *Wolbachia* and 53 min 52 sec on caterpillars injected twice with *Wolbachia* (Fig 7B).

## Discussion

In this study we show that parasitism by *C. glomerata* leads to a significant alteration of the odours of its host caterpillar *P. brassicae*, in which five VOCs are considered to be induced after parasitism. This aligns with changes in the internal and external bacterial communities, hinting at a relationship between caterpillars' body odours and their microbiome. Moreover, hyperparasitoids preferred the odours of starved, parasitised caterpillars (Cg-ST) over odours from starved then external microbiome disrupted parasitised caterpillars (Cg-ST+EMD), indicating a role of bacteria and/or their volatiles in the host-location of hyperparasitoids. The role of the internal microbiome was confirmed by the enhanced attraction of the hyperparasitoid to unparasitised caterpillars injected with *Wolbachia* over mock-injected caterpillars.

Analysis of the caterpillar headspace composition identified some consistent patterns in alterations of parasitised caterpillar odours regardless of whether caterpillars were starved (ST) or starved then external microbiome disrupted (ST+EMD). This indicates that the external microbiome and transient internal microbiome do not play a direct role in the production of characteristic VOCs. We found a significant association between parasitism and the caterpillar microbiome in the same sets of caterpillars. Both internal and external bacterial communities

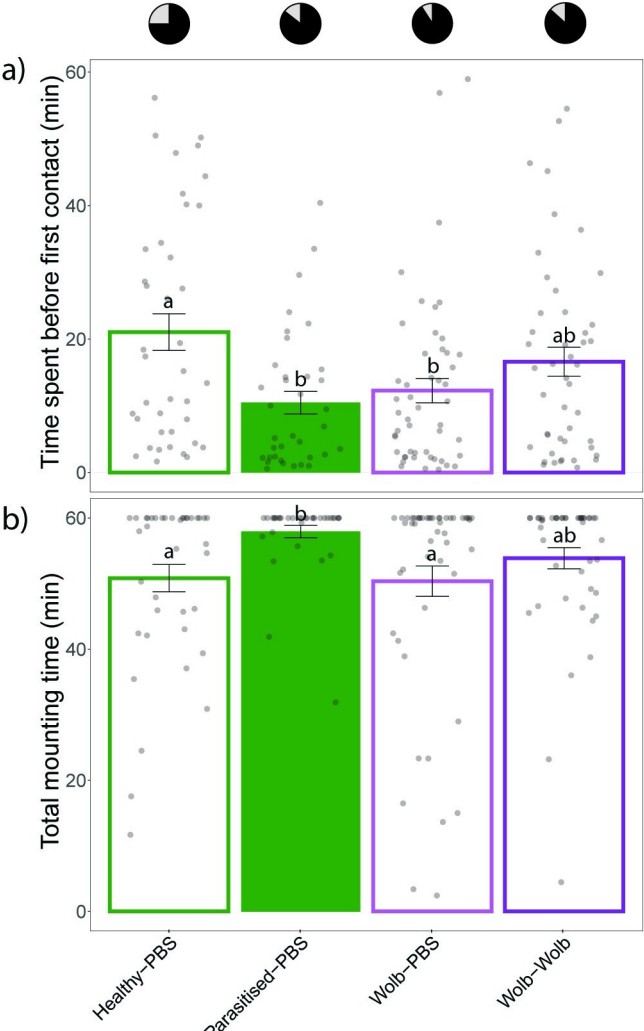

**Fig 7. Results of the no-choice assay to assess the role of *Wolbachia* in hyperparasitoid host location.** Bar plots showing (a) the average time (±SE) until first contact of *B. galactopus* with the caterpillar host *P. brassicae*, and (b) total mounting time (±SE) of *B. galactopus* on the caterpillars within 1 h after first contact was made. Hyperparasitoids were subjected to four treatments, including unparasitised caterpillars (Unparasitised-PBS; $n = 39$), caterpillars parasitised by *C. glomerata* (Parasitised-PBS; $n = 36$), caterpillars injected once with *Wolbachia* (Wolb-PBS; $n = 50$), and caterpillars injected twice with *Wolbachia* (Wolb-Wolb; $n = 46$). Statistical differences among treatments are indicated with different letters based on Kruskal-Wallis tests with Dunn's test for multiple comparisons including a Hochberg correction. Pie charts show the percentage of responding (black) and non-responding (grey) hyperparasitoids.

had a significant correlation with the caterpillar odour profile and we found nine bacterial zOTUs to be correlated with the VOCs produced by caterpillars. These bacteria were especially abundant in the internal microbiome, where the intracellular, resident insect symbiont *Wolbachia* sp. was only found in parasitised caterpillars.

Experimental injection of *Wolbachia* into unparasitised *P. brassicae* caterpillars had a significant effect on hyperparasitoid behaviour. Specifically, hyperparasitoids more rapidly made first contact with *Wolbachia*-injected caterpillars, unparasitized caterpillars compared to mock-injected unparasitized caterpillars. This response to *Wolbachia*-injected caterpillars was similar to parasitised caterpillars. In contrast, total mounting time was not affected by *Wolbachia*-injection, suggesting that this bacterium plays an important role in attracting the

hyperparasitoids, but does not arrest the hyperparasitoid on the caterpillar. Most likely, other cues linked to the presence of parasitoid larvae, like mechanosensory cues, are necessary for hyperparasitoids to identify host presence in caterpillars [38].

The VOCs (Z)-3-hepten-1-ol, 3-octanol, 3-pentanone, 3-methyl-3-buten-1-ol and 3-octanone were characteristic for parasitised *P. brassicae* caterpillars in the L5 stage and can potentially be used by its hyperparasitoids to reliably locate their host. Most of these characteristic compounds are known to be produced as (alarm) pheromones by other insects [44–47], but little is known about (Z)-3-hepten-1-ol. A parasitic fly is known to use an ant alarm pheromone, consisting of 3-octanone, 3-octanol and 3-nonanone, to locate its host [45]. 3-Methyl-3-buten-1-ol is an olfactory-active compound to a coccinellid predator to locate and perceive its aphid prey [48]. Additionally, a blend of 3-octanone and 3-octanol is released upon death of a social termite, allowing conspecifics to recycle the nutrients from the corpse before decomposition starts with higher chances of pathogen infection [46]. These compounds may also be a signal of impending death for parasitised *P. brassicae*, which benefits conspecifics in avoiding natural enemies but also provides its hyperparasitoid with a reliable cue to locate its host. Interestingly, none of the characteristic VOCs for parasitised caterpillars were detected in an earlier study on body odours of the closely related *P. rapae*, parasitised by the same parasitoid species *C. glomerata* [38]. It is therefore less likely that characteristic compounds are directly produced by the parasitoid larvae growing inside the caterpillar. In *P. rapae*, 2,3-butanedione was the only VOC found at higher levels in parasitised caterpillars [38], but it had a low impact on the separation between treatments in our current study (VIP score < 1). This indicates that the alteration of organismal odours after parasitism is species-specific and not overruled by parasitism, as known for plant odours induced by parasitised caterpillars [49]. Nearly all VOCs identified here as frass-related were previously found in caterpillar frass of both *P. rapae* and *P. brassicae* (excluding only 1-phenylethanol) [23], confirming their origin and indicating that frass odour is more dependent on food plant than caterpillar species as both caterpillars were reared on Brussels sprouts plants.

Our data show that the internal and external microbiome are altered after parasitism and have a significant correlation with the caterpillar odour profile. This is in line with our expectation that alterations in the microbiome coincide with changes in caterpillar odours through the production of mVOCs. Among their presence in various taxa, four out of five characteristic VOCs for parasitised caterpillars are known to be mVOCs according to the mVOC 3.0 database [50]. 3-Pentanone, 3-methyl-3-buten-1-ol and 3-octanone are produced by a variety of bacteria and fungi, whereas 3-octanol is only reported to be produced by fungi [50–52]. We did not find evidence that these compounds are directly produced by the selected bacteria in this study. Furthermore, it cannot be excluded that fungi play a role in the production of caterpillar odours, but we did not study fungal communities, as they occur in low abundance in association to *P. brassicae* [41]. To improve our understanding of the relationship between selected bacteria, their produced mVOCs and trophic interactions, future studies should consider to perform fermentations with these bacteria, followed by collection and characterisation of their headspace. This can be combined with behavioural assays of (hyper)parasitoids to evaluate their effect on trophic interactions [14,20].

Our results indicate that the internal bacterial communities were not substantially affected by starvation and starvation followed by external microbiome disruption treatments. Thus resident internal bacteria are most likely to be responsible for the difference between parasitised and unparasitised caterpillars. Due to our sampling method these bacteria can originate from the gut, haemolymph or any other internal tissues, including parasitoid larvae present in parasitised caterpillars. In accordance with previous studies, we found that the internal microbiome of *P. brassicae* mainly consisted of *Enterococcus* sp. and Enterobacteriaceae [41,53,54].

Among selected bacteria which contribute most to the variation in odour composition *Wolbachia* sp. and *Sphingomonas* sp. were most abundant and both are present in the resident internal microbiome. *Sphingomonas* sp. was especially abundant in unparasitised caterpillars and *Wolbachia* sp. was only present after parasitism confirming previous data [41]. *Sphingomonas* is known to be a pathogen-repressing- and plant-growth-promoting rhizobacterium. Its genome indicates that it has properties to produce terpenoids and degrade aromatic compounds [55]. However, both terpenoids and aromatic compounds are not involved in differences between unparasitised and parasitised caterpillars. The performed amplicon sequencing does not allow us to explore the genome of the *Sphingomonas* bacterium detected into more depth. Isolation and culturing of this bacterium and performing fermentations and characterisation of the produced mVOCs are needed to further investigate its biosynthetic capacities.

The exclusive presence of *Wolbachia* in parasitised caterpillars can be explained by the fact that it is transferred upon oviposition from the parasitoid into the caterpillar. Moreover, it has been shown in high relative abundance in parasitoid larvae [29,41]. *Wolbachia* is known to be involved in susceptibility to (hyper)parasitoids [56,57], but the mechanism behind this and its relationship to mVOCs and organismal odours are still unknown [56]. Our experiments involving micro-injections with *Wolbachia* confirm the hypothesis that *Wolbachia* is involved in hyperparasitoid host location, but further research is needed to fully understand its contribution.

*Wolbachia* is mostly known for its ability to alter the reproductive system, behaviour and metabolism of its insect host [58]. Moreover, it has been demonstrated that *Wolbachia*'s presence influences cuticular hydrocarbons and odour-mediated mate preference in a terrestrial isopod [59]. It can be hypothesised that the presence of *Wolbachia* in the internal microbiome after parasitism alters the caterpillar's odour through various mechanisms, which in turn may lead to changes in hyperparasitoid behaviour. For example, changes in the caterpillar's odour profile could be caused (i) by directly producing mVOCs which diffuse to the outside of caterpillars through the spiracles, (ii) by impacting caterpillar metabolism, thus providing other bacteria with a different set of metabolites, which are fermented to signature volatile compounds, or (iii) by indirectly affecting the bacterial communities, and thus the production of mVOCs through its presence. Its obligate intracellular lifestyle makes it difficult to cultivate the bacterium and carry out fermentations or other experiments [14,20]. However, insect cell-lines could provide a solution to characterise *Wolbachia*'s volatile metabolites [60].

Behavioural assays indicate that not frass odours, but the presence of the external microbiome on caterpillars are associated with the preference of *B. galactopus* (Figs 6 and S6). The external caterpillar microbiome is affected by parasitism, but the selected zOTUs on the external microbiome are low in relative abundance, making them poor candidates to produce a substantial amount of mVOCs. This is confirmed by negligible differences in body odours between starved (ST) and, starved and external microbiome disrupted caterpillars (ST+W +EMD) (S3 and S4 Figs). The diversity of the skin microbiome and presence of individual bacteria can also play a role in attracting insects [13,14,16,19,20,61]. Especially parasitised caterpillars undergoing the external microbiome disruption treatment after starvation had a lower diversity of bacteria on their skin, which could explain their reduced attractiveness to the hyperparasitoid (S7 Fig).

The hyperparasitoid *B. galactopus* was unable to discriminate between starved parasitised and starved unparasitised *P. brassicae* caterpillars based on their odours, as opposed to previous findings in *P. rapae*. Possibly, the starvation treatment applied in our current study with *P. brassicae* reduced the differences between parasitised and unparasitised caterpillars on which hyperparasitoids rely. These manipulations were not performed in the previous study on *P. rapae* [38]. The compound 2,3-butadione could play an important role in the ability of

hyperparasitoids to discriminate between healthy and parasitised caterpillars as shown for *P. rapae* [38]. The presence of this compound is not strongly affected by parasitism in *P. brassicae*, making it more difficult to differentiate between parasitised and unparasitised *P. brassicae* for this hyperparasitoid when relying on this cue. We cannot rule out that in addition to volatiles, also contact cues are used to determine parasitism status by the hyperparasitoid. Cuticular hydrocarbons are known to be important for aphid hyperparasitoids to locate their host [62].

We found that both the external and internal microbiome of *P. brassicae* caterpillars are altered after parasitism by *C. glomerata*. In contrast, our prior study indicated that *C. glomerata* parasitism only affected the internal microbiome of *P. brassicae* [47]. Furthermore, in our previous study we found a reduction in the diversity of bacterial communities for both internal and external microbiomes after parasitism, whereas our current data identified relatively weak effects. Both studies are on the same study system, but previous findings [7,63–65] suggest that habitat is one of the strongest factors influencing the external microbiome of insects, outweighing the effects of parasitism due to differences in day, weather, environment and/or host plant [41]. As a result, parasitism status played an insignificant role in explaining variation in the external microbiome. To further investigate differences caused by parasitism, our current study pooled the microbiomes of groups of caterpillars and only examined one habitat (lab strain). This allowed us to identify that the external microbiome is altered alongside the internal microbiome. The difference in responses of Shannon diversity can most likely be attributed to the studied lab strain with an overall less diverse bacterial community [41] and to the higher resolution with more statistical power in this study.

In conclusion, our findings provide evidence that parasitism-induced changes in the caterpillar microbiome are responsible for alterations of its odours. For hyperparasitoids, the presence of the skin microbiome is more important than frass odours to locate its host. This is likely mediated by differences in diversity of the external microbiome as well as differentially abundant bacteria between parasitised and unparasitised caterpillars. We identified five VOCs related to parasitism, but did not confirm causality between specific bacteria and these caterpillar odours. The intracellular bacterium *Wolbachia* sp., which was only found in the internal microbiome of parasitised caterpillars, is the strongest candidate to cause the production of characteristic mVOCs for parasitised caterpillars although the mechanisms are still unclear. It may produce mVOCs directly by itself, but can also affect the bacterial community or the presence of metabolites. Micro-injections of caterpillars with *Wolbachia* confirmed that *Wolbachia* is involved in attraction of hyperparasitoids, but could not fully explain their behaviour. Further research is needed to elucidate the role of this intracellular bacterium and its products in the interaction between the host microbiome and its odours, which can be exploited by hyperparasitoids to locate their host. The identified compounds related to parasitised caterpillars can potentially be employed in optimizing biological control that is hampered by hyperparasitoids reducing the population of parasitoids that are used in biological control of herbivorous pests [66]. The bacterial odours (or synthetic counterparts) may be used in traps or a push-pull system to reduce hyperparasitoid populations in cropping systems [67]. However, this would first require a further evaluation of the attractiveness of these compounds on hyperparasitoids and non-target organisms.

## Materials and methods

### Insects and rearing

**Used insects.**    The larval stage (caterpillar) of the large cabbage white, *P. brassicae*, was selected as focal study object. Its parasitoid *C. glomerata* was included because it is known to

influence the caterpillars' microbiome after parasitism [41]. To elucidate the role of caterpillar odours in the host-location of higher trophic levels we used the hyperparasitoid *B. galactopus*. This hyperparasitoid is known to use caterpillar body odours during host location [38]. *Pieris brassicae* is an important oligophagous pest species whose larvae feed on many members of the Brassicaceae family such as cabbage, cauliflower, Brussels sprouts and rapeseed. In nature, *Pieris* spp. are attacked by various natural enemies, including highly specialised parasitoid wasps, known to have host-manipulating traits [29]. *Cotesia glomerata* is a gregarious koinobiont parasitoid, which parasitises a wide range of *Pieris* spp., with *P. brassicae* and *P. rapae* as its main hosts. The hyperparasitoid *B. galactopus* is widely distributed all around Europe and has been reported as one of the most common hyperparasitoids in the *Brassica oleracea*-associated insect community with a wide host range that includes braconid and ichneumonid parasitoids [38,68]. It is a true hyperparasitoid that parasitises the larvae of parasitoids that are concealed in the herbivore body [40].

**Insect rearing conditions.** Experiments were performed using lab cultures of *P. brassicae*, *C. glomerata* and *B. galactopus*, which were originally collected from agricultural fields in the near vicinity of Wageningen University & Research, the Netherlands. *Pieris brassicae* was reared in cages and maintained on Brussels sprouts (*B. oleracea* L. var. *gemmifera* cv. Cyrus) in greenhouse conditions (± 21°C, 25–35% RH, 16:8 h L:D). In a different greenhouse compartment with the same conditions, *C. glomerata* adult wasps were provided with first instar (L1) *P. brassicae* caterpillars and allowed to oviposit for 10–15 min. After oviposition, parasitised caterpillars were reared under the same conditions as unparasitised caterpillars. Parasitoid cocoons were collected a few days after they egressed from the caterpillar and placed in petri-dishes (90 mm) until emergence. Adult parasitoids were kept in a fine mesh cage and provided with honey and water. The hyperparasitoid *B. galactopus* was reared on *P. brassicae* caterpillars that had been parasitised by *C. glomerata*. To this end, to a Petri dish (90 mm) with seven parasitised L5 caterpillars, 15–20 *B. galactopus* adults were added. After egression of the parasitoid larvae from the caterpillar, the cocoons containing hyperparasitised *C. glomerata* were placed in a Petri dish (90 mm) in a climate chamber (22 ± 0.5°C, 50–70% RH, and 16:8 h L:D) until adult *B. galactopus* emerged. Adult *B. galactopus* were kept in a rearing cage at room temperature, under natural light conditions (± 22°C, 35–45% RH) and were supplied with 5% honey-water.

## Microbiome manipulation treatments

Previous research has shown that parasitism by *C. glomerata* alters the internal microbiome of *P. brassicae* caterpillars [41]. Therefore, half of the caterpillars studied were subjected to *C. glomerata* parasitism, while the other half remained unparasitised. When caterpillars had reached the fifth instar (L5), caterpillars were collected and subjected to different treatments. Each group of parasitised and unparasitised caterpillars was subjected to either (i) starvation (ST), (ii) starvation followed by a washing treatment to disrupt the external microbiome (ST+EMD) or (iii) were left untreated. Starvation was expected to affect the gut microbiome by exclusion of transient bacteria, but not the external microbiome. In addition, starvation was aimed at limiting the presence and effect of frass-related mVOCs. The starvation followed by external microbiome disruption was expected to affect the external microbiome in addition to the internal microbiome affected by the starvation treatment. Caterpillars undergoing starvation (ST and ST+EMD treatments) were starved for 24 hours by placing groups of ten caterpillars in a sterile plastic container with tissue paper (to absorb frass and moisture) during which they could empty their gut. Then those subjected to external microbiome disruption after starvation were washed three times with a 0.05% sodium hypochlorite solution followed by rinsing with

sterile water. We used a 0.05% sodium hypochlorite solution to disrupt the composition of the external microbiome of the caterpillars while preventing any mortality. This disruption reduces the absolute and relative abundance of most external microorganisms as a consequence of the different sensitivity to sodium hypochlorite and the differences in the adhesive capacity of bacteria [69,70]. Higher concentrations of sodium hypochlorite were proven to be more effective at removing bacteria but were too invasive for caterpillars. Only starved caterpillars were subjected to the external microbiome disruption treatment to limit excessive contaminations with bodily products (frass and regurgitate). Untreated caterpillars were also placed in groups of ten caterpillars in a sterile plastic container with tissue paper, but were supplied with fresh Brussels sprouts leaves and continued to feed, while other treatments were starved. Boxes with caterpillars had a pierced lid and were stored for 24 h at room temperature (± 22°C, 35–45% RH). Groups of eight caterpillars were subjected to headspace collection, followed by microbiome collection. In total every treatment was replicated ten times.

### Caterpillar odour profile analysis

**Headspace collection of caterpillar odours.** To characterize the odour profiles of the starved, starved then external microbiome disrupted or untreated parasitised and unparasitised caterpillars, headspace samples were collected from ten cohorts of eight caterpillars per treatment, resulting in a total of 60 samples. However, one sample was (Sample 11, Pb-ST treatment) an extreme outlier based on the RDA analysis and was hence removed from the dataset. We followed the methodology described by [38] in which caterpillar headspace samples were collected in 500 mL glass jars sealed with a Viton-lined glass lid with an inlet and outlet for (clean) air. In our setup, each glass jar contained eight L5 *P. brassicae* caterpillars from the same treatment (Fig 1). A continuous flow of clean, synthetic air (Air Synthetic 4.0 Monitoring from Linde Gas, Schiedam, The Netherlands) was used as a carrier of volatiles at a flow rate of 100 mL min$^{-1}$. Volatiles emitted by the caterpillars were trapped by drawing air out of the glass jar at a suction rate of 100 mL min$^{-1}$ through a stainless-steel tube filled with 200 mg Tenax TA (20/35 mesh; CAMSCO, Houston, TX, USA) for 2 h. To prevent caterpillars from moving upward and blocking the air in- or outlets, each collection jar contained a restriction device made from a stainless-steel dome sieve (Cuisine elegance, Ø = 8.5 cm, mesh size = 12 mm) on a basis of aluminium foil (Fig 1). This device physically divided the setup into two to restrict caterpillar movement, but air could move freely through both parts. Directly after headspace collection, the Tenax TA cartridges were dry-purged under a stream of helium (50 ml min$^{-1}$) for 10 min at room temperature (21 ± 2°C) to remove moisture before storage. In order to prevent any contribution from the collection set-up, the adsorbent material and the analytical system, we routinely trapped volatiles from empty jars containing a restriction device and included these as background samples.

**Separation and detection of volatile compounds.** The collected volatiles were thermally released from the Tenax TA adsorbent using an Ultra 50:50 thermal desorption unit (Markes, Llantrisant, Glamorgan, UK) at 250°C for 10 min under a helium flow of 20 mL min$^{-1}$. Simultaneously the volatiles were re-collected in a thermally cooled universal solvent trap: Unity (Markes) at 0°C. Once the desorption process was completed, volatile compounds were released from the cold trap by ballistic heating at 40°C s$^{-1}$ to 280°C. This was then kept for 10 min, while all the volatiles were transferred to a ZB-5MS analytical column (30 mL x 0.25 mm ID x 1 mm F.T.) with 10 m built-in guard column (Phenomenex, Torrance, CA, USA), placed inside the oven of a Thermo Trace GC Ultra (Thermo Fisher Scientific, Waltham, MA, USA), for further separation of the volatiles. The gas chromatograph oven temperature was initially

held at 40˚C for 2 min and was immediately raised at 6˚C min$^{-1}$ to a final temperature of 280˚C, where it was kept for 4 min under a constant helium flow of 1 mL min$^{-1}$.

A Thermo Trace DSQ quadrupole mass spectrometer (Thermo Fisher Scientific) coupled to the gas chromatograph was operated in an electron impact ionisation (EI) mode at 70 eV in a full scan with a mass range of 35–400 amu at 4.70 scans s$^{-1}$. The MS transfer line and ion source were set at 275 and 250˚C, respectively. Automated baseline correction, peak selection (S/N > 3) and alignments of all extracted mass signals of the raw data were processed following an untargeted metabolomic workflow using MetAlign. This software uses the height of the peak after baseline correction to avoid problems with peak overlap, varying baselines, and peak tailing among others [71]. This produces detailed information on the relative abundance of mass signals representing the available metabolites. Next, the reconstruction of the extracted mass features into potential compounds was done using the MSClust software through data reduction by means of unsupervised clustering and extraction of putative metabolite mass spectra [72]. Tentative identification of volatile metabolites was based on comparison of the reconstructed mass spectra with those in the NIST 2008 and Wageningen Mass Spectral Database of Natural Products MS libraries, as well as experimentally obtained linear retention indices (LRI, Table 1).

**Statistical analysis of volatile data.**   The volatile emission data as peak heights were imported to SIMCA-P 17 statistical software (Umetrics, Umea, Sweden), followed by log-transformation, mean-centering and unit-variance scaling before subjecting the data to multivariate data analysis. Supervised orthogonal partial least squares-discriminant analysis (OPLS-DA) and unsupervised principal component analysis (PCA) statistical models were used as a tool to compare and correlate treatment groups. Whereas PCA was used when no significant model for OPLS-DA could be obtained. The results were visualised as score plots revealing the sample structure according to the model components, and loading plots showing the contribution of variables to the components as well as the relationship among the variables. R2 and Q2 metrics, which describe the explained variation within the data set and the predictability of the model, were calculated based on the averages of the sevenfold cross-validation. R2 and Q2 values range between 0 and 1, and the closer these metrics are to 1, the higher the variance explained by the model and the more reliable the predictive power of the model. Permutation and CV-ANOVA were used to validate models.

After identifying compounds with a variable importance in the projection (VIP) score of 1 or higher as potentially relevant, we subjected these to further analyses using a Kruskal-Wallis test followed by Dunn's test for multiple comparisons with a Bonferroni correction to determine significant differences between treatments. This was done using the dunn.test command in R-studio (dunn.test package) [73,74].

## Caterpillar microbiome analysis

**Microbiome collection and processing.**   The external and internal microbiome were immediately sampled after volatile collection, as described previously [41,64] (Fig 1). Briefly, to collect the external microbiome, caterpillars were put individually in a 2 mL microcentrifuge tube containing 1 mL of phosphate-buffered saline with 0.01% Tween80 (PBS-T), and vortexed for 20 seconds. The resulting solution then reflected the external microbiome of each individual. To remove any residual microorganisms, caterpillars were transferred into a new tube with 1 mL sodium hypochlorite (2.5%) and vortexed again for 20 seconds, followed by a final washing step in PBS-T. Each caterpillar was then dissected under sterile conditions to confirm presence or absence of *C. glomerata* larvae. Next, samples were pooled according to the same groups of eight caterpillars that were used for the volatile collection. To pool the

external microbiomes, the washing solution of the first caterpillar was centrifuged ($13000 \times g$, 10 min, room temperature), followed by removal of the supernatant. Subsequently, the washing solution of the second caterpillar of the same group was added to the same tube and centrifuged, after which the supernatant was removed again. This process was carried out until each of the eight caterpillars was included. The resulting pellet was resuspended in 500 µL nuclease free water and subsequently used for DNA extraction. For the internal samples, the dissected caterpillars were aseptically combined in a sterile 100 mL glass beaker and mixed under sterile conditions with an immersion homogeniser (RIVIERA&BAR, PPM530) at maximum speed for 20 s. The obtained mixture was then transferred into a 15 mL Falcon tube, vortexed and used for DNA extraction.

**Microbial community characterisation.** For each pooled sample, genomic DNA was extracted from 500 µL sample material using the PowerPro Soil Kit (Qiagen, Hilden, Germany) following the manufacturer's instructions, with one modification: in the second step of the protocol the use of a vortex adapter was replaced by two cycles of 30 s in the Bead Ruptor Elite at a speed of 5.5 m s$^{-1}$. To confirm the absence of contamination from the kit reagents', two negative controls were included by replacing the sample with 500 µL sterile, DNA-free water. Next, for each sample the hypervariable V4 region of the bacterial 16S rRNA gene was amplified using Illumina-barcoded versions of primers 515F and 806R [75], designed according to a dual index sequencing strategy [76] (Table C in S1 Supporting Information). Additionally, four negative controls in which template DNA was replaced by DNA-free water were included, as well as a DNA mock community sample, composed of a number of bacterial species that likely occur in or on insects [41] (Table D in S1 Supporting Information). PCR amplification, library preparation, sequencing and bioinformatics analysis were performed as described previously [41]. Sequences were classified into zero-radius operational taxonomic units (zOTUs, also known as amplicon sequence variants (ASVs) [77,78]), enabling to resolve sequence differences by as little as a single base pair. Further, to remove potential contaminants, the data set was decontaminated in R (v4.0.2) using microDecon (v1.2.0) [79] based on zOTU prevalence in the investigated samples versus the mean of the four PCR controls [74,80]. Additionally, in accordance with the results obtained for the mock community, zOTUs occurring below a 0.3% relative abundance threshold per sample were discarded from further analysis. Finally, the number of sequences was rarefied to 1,000 sequences per sample, representing the least number of high-quality sequences obtained per sample. The taxonomic origin of each zOTU was determined with the SINTAX algorithm as implemented in USEARCH based on the SILVA Living Tree Project v1.23 (Table E in S1 Supporting Information). The identity of the most important zOTUs was further verified with a BLAST search in GenBank against type materials. When no significant similarity was found with type materials, the BLAST search was performed against entire GenBank. The negative DNA extraction samples showed no visible band after the amplification, therefore no contaminants were present in the DNA extractions. Analysis of the mock community demonstrated that only the expected taxa were found, indicating that the experimental conditions were met to achieve robust data. The sequences obtained in this study were deposited in the Sequence Read Archive (SRA) at NCBI under Bioproject PRJNA878850.

**Statistics on microbiome data.** For each sample, a rarefaction curve was generated using the Phyloseq package in R showing the number of observed zOTUs as a function of the number of sequences [74,81]. For all samples, rarefaction curves approached saturation, indicating that our sequencing depth was sufficient to cover the microbial diversity (S7 Fig). Phyloseq package in R was used as well to calculate observed zOTU richness and Shannon diversity. A three-way analysis of variance (ANOVA) was used to assess whether health status (parasitised or unparasitised), origin (external or internal microbiome) and treatment affected zOTU

richness and Shannon diversity. All two-way interactions and three-way interactions were included in the model as well. A further analysis of variance was conducted on external and internal samples separately. The bacterial community composition was visualised by non-metric multidimensional scaling (NMDS) using the Bray-Curtis coefficient as distance measure in the R software package Vegan [82]. The coefficient was based on the Hellinger transformed relative abundance data of the observed bacteria in each sample. To test whether bacterial communities differed between health status, origin and treatments, the "adonis" function in the software package Vegan [82] was used to perform a permutational analysis of variance (PERMANOVA) [83]. All factors and their interactions were included as fixed factors in the analysis. Significance was tested using 1,000 permutations.

### Forward selection RDA to link microbiome with volatile data

To investigate whether the volatile composition of parasitised and unparasitised caterpillars was significantly related to variation in microbiome, redundancy analyses using the rda function in Vegan [80] were used. In these analyses, log transformed peak heights were used as dependent variables, whereas zOTU relative abundance data were used as independent data. A forward selection procedure using the ordistep function in Vegan was applied to retain those zOTUs that had a significant impact on volatile composition. Significance of the final model was tested using the anova.cca function in Vegan [80] using 1000 permutations. Finally, triplots were constructed, which allow to visualise the relationship between samples, volatiles and the selected zOTUs. Data were analysed separately for the external and internal microbiome.

### Behavioural assays with hyperparasitoids

**Hyperparasitoid preference for caterpillar odours.** The response of mated *B. galactopus* females to the odours of differently treated unparasitised or parasitised *P. brassicae* was tested in a Y-tube olfactometer. A similar setup was successfully used to study the preference of the parasitoid wasp *Nasonia vitripennis* (Hymenoptera: Pteromalidae) to odours of conspecifics [84]. A glass Y-tube with an 8.5 cm stem, two 5 cm arms and a diameter of 0.9 cm (Fig 8A) was tilted upward on a white wooden board with an angle of 40˚ towards a single light source (TL-D 58 W, Phillips, the Netherlands) (Fig 8B). A fine gauze mesh was placed over each end of the Y-tube to create a physical barrier, then a 5 mL transparent pipette tip containing the test caterpillars was mounted on each end of the Y-tube. To test *B. galactopus'* preference for *C. glomerata* odours, two fifth instar *P. brassicae* caterpillars were placed in a pipette tip (Fig 8A). For the preference of *H. ebeninus* odours this were four third instar *P. brassicae* caterpillars because *H. ebeninus* restricts the growth of its host [85]. A charcoal-filtered airflow of 500 mL min$^{-1}$ was split and led through both arms, starting at the narrow end of the pipette tip and was sealed with Teflon tape (Fig 8A).

Prior to each set of choices, caterpillar odours were allowed to diffuse for 15 min from the pipette tips containing caterpillars into the arms of the Y-tube olfactometer. If present, frass droppings were removed prior to testing. Tests started when one *B. galactopus* female was released at the base of the Y-tube from a glass vial. A choice was noted when the hyperparasitoid passed a drawn line at 1 cm from the end of either arm for 15 seconds. If 10 min had passed without the hyperparasitoid making a choice, this was recorded as a no-choice and excluded from statistical analysis. Each female was used only once and a total of 10 females per day was used per tested combination. After every 5 tested females the Y-tube was turned by 180 degrees to compensate for any unforeseen asymmetry bias. We subjected unparasitised and parasitised *P. brassicae* to the same treatments as mentioned before. Untreated caterpillars were omitted from choice-assays to prevent frass production from contaminating the setup

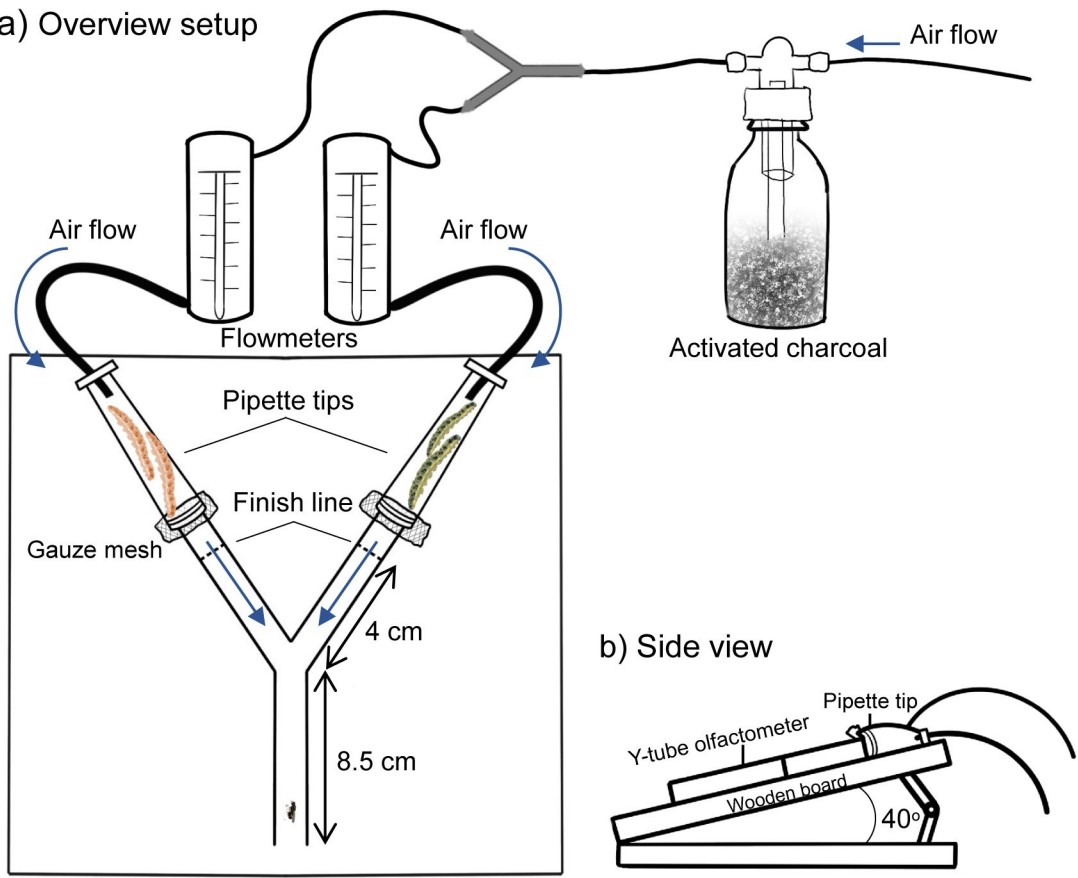

**Fig 8. Y-tube olfactometer setup.** a) The setup consisted of a Y-tube with an 8.5 cm-long stem, two 5-cm-long arms (angle between both arms of 65˚) and a diameter of 0.9 cm. A fine gauze mesh was placed over each end of the Y-tube to create a physical barrier, then a 5 mL transparent pipette tip containing caterpillars was mounted on each end of the Y-tube. A charcoal filtered airflow of 500 mL min⁻¹ was split, continuously measured with flowmeters and led through both arms, sealed with Teflon tape. Hyperparasitoid females were released at the basis of the Y-tube and tests concluded once they passed the finish line for at least 15 seconds. b) The setup was mounted on a wooden board and tilted upward with an angle of 40˚ towards a single light source.

and overruling the effect of caterpillar body odours. The preference of *B. galactopus* for caterpillar odours was analysed using two-tailed binomial tests with the binom.test command in R-studio [74].

### Hyperparasitoid preference for caterpillar frass odours

We used a two-chamber olfactometer setup [38] to assess the attractiveness of frass odours from (parasitised) caterpillars to the true hyperparasitoid *B. galactopus*. Three treatments were included, an empty chamber (E), a chamber containing a sample of 10 pellets of frass produced by an unparasitised L5 *P. brassicae* caterpillar (Pb) or by an L5 *P. brassicae* caterpillar parasitised by *C. glomerata* (Cg). For each treatment combination, 71 replicates in total were included. In order to maintain the frass equally fresh, a filter paper moisturised with sterile Milli-Q water was placed at the bottom of all chambers, including empty ones. Five mated female *B. galactopus* wasps were released inside the experimental arena containing the two chambers. After exactly one hour, the number of wasps present in each chamber was assessed for each combination. When no wasps were found inside any of the two chambers at the end

of the bioassay, the cohort was considered unresponsive and was excluded from the analysis. Bioassays of the three treatment combinations were always performed during the same time, in order to account for variation caused by hour and day. The bioassays were conducted between 10.00 and 16.00h. After each bioassay, the chambers and parts of the set up in contact with frass were wiped with 70% ethanol, dried with tissue paper and the whole set up was flushed with high-pressure clean air. A cohort of 5 wasps was considered as a replicate. Thus for each of the combinations we ended up with two distributions of scores from 0 to 5. We compared the two non-parametric distributions and their median with Mann-Whitney U tests separately for each treatment combination in R-studio using the wilcox.test command (unpaired) [74].

## Hyperparasitoid behavioural responses to caterpillars injected with *Wolbachia*

Caterpillars were injected according to the method described in [86] with modifications. *Wolbachia* was extracted from maturing *C. glomerata* larvae instead of adult parasitoids. Because *Wolbachia* in adult parasitoids co-occurs with abundant polydnaviruses in the calyx fluid of the parasitoid ovaries, extraction of *Wolbachia* from the calyx would lead to contamination with polydnavirus that is known to affect the caterpillar's phenotype [87]. Previous research has shown that the internal microbiome of *C. glomerata* larvae is dominated by *Wolbachia*, up to 100% relative abundance [41] and do not contain polydnavirus particles. After anaesthesia, L5 *P. brassicae* caterpillars parasitised by *C. glomerata* were aseptically dissected and the parasitoid larvae were collected and surface-sterilized as described in [41]. Parasitoid larvae were crushed in a centrifuge tube with a small pestle in 20 μL sterile PBS and subsequently homogenized in 800 μL PBS with two glass beads (2 mm diameter) using a TissueLyser II (Qiagen, Hilden, Germany; 3 min at 30 Hz). Following a 10-min centrifugation at 2500 g, the supernatant was collected and filtered through a Puradisc 13 glass microfiber syringe filter (pore size: 2.7 μm; Whatman, Amersham, UK). The filtered solution was then centrifuged at 18500 g for 5 min to pellet the bacteria. The cell pellet was re-suspended in 10 μL sterile PBS and used for the injections. Parasitoid larvae were processed in groups of 20 individuals to obtain *Wolbachia* densities similar to those found in 10 ovary pairs of *C. glomerata*. The concentration of *Wolbachia* per ovary pair was determined at $2 \times 10^2$ 16S rRNA gene copies per μL DNA by qPCR; for primers and PCR conditions, see [88]. Caterpillars were anaesthetized with $CO_2$ and cell suspensions were injected into *P. brassicae* caterpillars (right behind the head) using a Femtojet microinjector (Eppendorf, Hamburg, Germany). Injections were performed with aliquots of 0.1 μL, assuming that 1/10 of calyx fluid is injected with the eggs during a parasitism bout [86].

The following treatments were included: (1) unparasitised caterpillars injected twice with PBS (0.1 μL) (Unparasitised-PBS; negative control); (2) parasitised caterpillars injected twice with PBS (Parasitised-PBS; positive control); (3) unparasitised caterpillars injected first with *Wolbachia* and then with PBS (Wolb-PBS); and (4) unparasitised caterpillars injected twice with *Wolbachia* (Wolb-Wolb). The first injection was performed on first instar caterpillars, while the second injection was performed the day before performing the behavioural assay with caterpillars at the 5th instar. Caterpillars were maintained in separate cages per treatment under the previous mentioned conditions until they were used in the behavioural assays.

Behavioural assays were performed when caterpillars had reached the fifth instar as described in [38]. One caterpillar and one mated naïve female of *B. galactopus* were placed in a clean 9 cm diameter Petri dish (1.9 cm height), after which the hyperparasitoid's behaviour was monitored. Over one hour, we recorded the time it took for *B. galactopus* to make first

contact and mount the caterpillar. Next, in the hour following mounting, the total time the hyperparasitoid spent on top of the caterpillar was recorded. If the hyperparasitoid did not mount on the caterpillar within the period of one hour, it was considered as a non-responder and removed from the analysis. Furthermore, to ensure that the positive control only consisted of parasitised caterpillars, parasitised individuals were dissected in order to confirm parasitism, and caterpillars without parasitoid larvae were discarded from the analysis. In total, the experiment was performed with 39 caterpillars for the unparasitised-PBS group, 36 for Parasitised-PBS, 50 for Wolb-PBS, and 46 for Wolb-Wolb. To investigate whether the time used to make contact and total mounting differed between treatment, a non-parametric Analysis of Variance was used. Subsequently, post-hoc comparisons using Dunn's test were performed to see which treatment combinations differed significantly from each other. P-values were adjusted for multiple comparisons using the Hochberg method.

## Supporting information

**S1 Fig. Overview of caterpillar-associated bacterial communities and volatiles of starved then external microbiome disrupted unparasitised- or *C. glomerata* parasitised caterpillars (*Pieris brassicae*).** a) NMDS ordination plots based on Bray–Curtis distances of Hellinger-transformed relative abundance data of the external (dermal) bacterial communities. b) NMDS ordination for the internal bacterial communites. c) OPLS-DA plot for the volatile blends of different groups of caterpillars. The Hotelling's T2 ellipse confines the confidence region (95%) of the score plot. d) The loading plot defining the contribution of each of the volatile compound to the separation of treatment groups. Volatiles compounds closer to a treatment in the plot are more correlated. For compound identity see Table 1. Abbreviations used: Cg-ST+EMD = starved then external microbiome disrupted parasitised caterpillars. Pb-ST+-EMD = starved then external microbiome disrupted unparasitised caterpillars. (TIF)

**S2 Fig. Overview of caterpillar-associated bacterial communities and volatiles of starved unparasitised- or *C. glomerata* parasitised caterpillars (*Pieris brassicae*).** a) NMDS ordination plots based on Bray–Curtis distances of Hellinger-transformed relative abundance data of the external (dermal) bacterial communities. b) NMDS ordination for the internal bacterial communites. c) OPLS-DA plot for the volatile blends of different groups of caterpillars. The Hotelling's T2 ellipse confines the confidence region (95%) of the score plot. d) The loading plot defining the contribution of each of the volatile compound to the orientation of sample groups in the score plot. Volatiles compounds closer to a treatment in the plot are more correlated. For compound identity see Table 1. Abbreviations used: Cg-ST = starved parasitised caterpillar. Pb-ST = starved unparasitised caterpillar. (TIF)

**S3 Fig. Overview of caterpillar-associated bacterial communities and volatiles of starved then external microbiome disrupted or only starved *C. glomerata* parasitised caterpillars (*Pieris brassicae*).** a) NMDS ordination plots based on Bray–Curtis distances of Hellinger-transformed relative abundance data of the external (dermal) bacterial communities. b) NMDS ordination for the internal bacterial communites. c) PCA plot for the volatile blends of different groups of caterpillars. The Hotelling's T2 ellipse confines the confidence region (95%) of the score plot. d) The loading plot defining the contribution of each of the volatile compound to the orientation of sample groups in the score plot. For compound identity see Table 1. Abbreviations used: Cg-ST = starved parasitised caterpillar; Cg-ST+EMD = starved

then external microbiome disrupted parasitised caterpillar.
(TIF)

**S4 Fig. Overview of caterpillar-associated bacterial communities and volatiles of starved then external microbiome disrupted or starved unparasitised caterpillars (*Pieris brassicae*).** a) NMDS ordination plots based on Bray–Curtis distances of Hellinger-transformed relative abundance data of the external (dermal) bacterial communities (stress = 0.132). b) NMDS ordination for the internal bacterial communites (stress = 0.115). c) PCA plot for the volatile blends of different groups of caterpillars. The Hotelling's T2 ellipse confines the confidence region (95%) of the score plot. d) The loading plot defining the contribution of each of the volatile compound to the first two principal components. For compound identity see Table 1. Abbreviations used: Pb-ST = starved unparasitised caterpillar; Pb-ST+EMD = starved then external microbiome disrupted unparasitised caterpillar.
(TIF)

**S5 Fig. *B. galactopus* preference for frass odours.** The number of *B. galactopus* found inside a chamber of the two-chamber olfactometer after one hour (out of 5 individuals). Chambers were either empty or contained fresh frass from L5 unparasitised or parasitised caterpillars. *N* = the number of replicates, for each replicate 5 *B. galactopus* were released in the olfactometer. *P* was calculated using a Mann-Whitney U-test.
(TIF)

**S6 Fig. Preference of hyperparasitoids for caterpillar body odours to healthy and parasitised caterpillars in the L3 stage. For this experiment caterpillars were parasitised by the parasitoid *Hyposoter ebeninus*.** Preference of the hyperparasitoid *Baryscapus galactopus* for caterpillar body odours was tested with Y-tube olfactometer tests. Tested combinations were selected according to the hypothesis that hyperparasitoids can use caterpillar body odours, which are (at least partially) determined by the external microbiome. Numbers between brackets indicate the number of wasps that made a choice within 10 min from the start of the experiment versus the total number of wasps tested. *** *P* <0.001; two-sided binomial test. Abbreviations used: AIR = clean air; HE = Untreated parasitised caterpillar; HE-ST = starved parasitised caterpillar; HE-ST+EMD = starved then external microbiome disrupted parasitised caterpillar. Pb = Untreated unparasitised caterpillar; Pb-ST = starved unparasitised caterpillar; Pb-ST+EMD = starved and external microbiome disrupted unparasitised caterpillar.
(TIF)

**S7 Fig. Overview of the diversity of caterpillar-associated bacterial communities of untreated, starved or starved then external microbiome disrupted *C. glomerata* parasitised or unparasitised caterpillars (*Pieris brassicae*).** a) The shannon diversity of the external microbiome b) The Shannon diversity of the internal microbiome.
(TIF)

**S8 Fig. Rarefaction curves for the different samples studied, based on the bacterial V4 dataset.** Rarefaction curves approached saturation, indicating that our sequencing depth was sufficient to cover the microbial diversity.
(TIF)

**S1 Supporting Information.  Table A: VIP scores for the volatile compounds in pairwise comparisons of untreated, starved then external microbiome disrupted or starved unparasitised- or *C. glomerata* parasitised caterpillars (*Pieris brassicae*).** Bold face VIP scores are higher than 1 and indicate the most influential VOC for separation of the mentioned treatments. Compounds and pairwise comparisons in green are correlated with the presence/

absence of frass. Compounds and pairwise comparisons in yellow are correlated with parasitism status. **Table B: Full overview of caterpillar-associated bacterial communities of untreated, starved then** external microbiome disrupted **or starved *C. glomerata* parasitised or unparasitised caterpillars (*P. brassicae*). Table C: Primer design and sample-specific barcodes. Table D: Composition of mock community. Table E: Identification of bacterial zero radius operational taxonomic units (ZOTUs) according to the Silva v1.23 database and distribution over the investigated samples**.
(XLSX)

## Acknowledgments

We thank Pieter Rouweler, Arnout Berendsen, the late Frans van Aggelen, and André Gidding for rearing *P. brassicae* and *C. glomerata*. Further, we are grateful to Bram Kamps and Maximilien Cuny for rearing *H. ebeninus*, Janneke Bloem and Filippo Guerra for assistance in the lab.

## Author Contributions

**Conceptualization:** Mitchel E. Bourne, Gabriele Gloder, Marijn Slingerland, Bart Lievens, Hans Jacquemyn, Marcel Dicke, Erik H. Poelman.

**Data curation:** Mitchel E. Bourne, Gabriele Gloder, Sam Crauwels, Bart Lievens.

**Formal analysis:** Mitchel E. Bourne, Gabriele Gloder, Berhane T. Weldegergis, Andrea Ceribelli, Sam Crauwels, Hans Jacquemyn.

**Funding acquisition:** Bart Lievens, Hans Jacquemyn, Marcel Dicke.

**Investigation:** Mitchel E. Bourne, Gabriele Gloder, Berhane T. Weldegergis, Marijn Slingerland, Andrea Ceribelli.

**Methodology:** Mitchel E. Bourne, Gabriele Gloder, Berhane T. Weldegergis, Marijn Slingerland, Andrea Ceribelli.

**Project administration:** Bart Lievens, Marcel Dicke.

**Resources:** Marcel Dicke.

**Software:** Sam Crauwels.

**Supervision:** Bart Lievens, Hans Jacquemyn, Marcel Dicke, Erik H. Poelman.

**Validation:** Mitchel E. Bourne, Berhane T. Weldegergis, Marcel Dicke, Erik H. Poelman.

**Visualization:** Mitchel E. Bourne, Gabriele Gloder, Berhane T. Weldegergis, Andrea Ceribelli, Hans Jacquemyn.

**Writing – original draft:** Mitchel E. Bourne, Gabriele Gloder, Erik H. Poelman.

**Writing – review & editing:** Mitchel E. Bourne, Gabriele Gloder, Berhane T. Weldegergis, Marijn Slingerland, Andrea Ceribelli, Sam Crauwels, Bart Lievens, Hans Jacquemyn, Marcel Dicke, Erik H. Poelman.

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
