## [Decision Letter · Decision Letter 0]

10 Dec 2022

Dear Bourne,

Thank you very much for submitting your manuscript "Parasitism causes changes in caterpillar odours and associated bacterial communities with consequences for host-location by a hyperparasitoid" for consideration at PLOS Pathogens. As with all papers reviewed by the journal, your manuscript was reviewed by members of the editorial board and by several independent reviewers. In light of the reviews (below this email), we would like to invite the resubmission of a significantly-revised version that takes into account the reviewers' comments.

Based upon my own reading of the manuscript and the consensus of the three reviewers, this is a very interesting and well-written study of the odors involved in parasitism by hyperparasitoids. The attempts of the authors to investigate the origin of odors that are attractive to hyperparasitoids was appreciated. The research topic is fascinating.

I agree with the reviewers that some substantial revisions are necessary. In the aggregate, the three reviewers outline several major issues that the authors will need to address, and I agree with these. I consider that the most crucial of these are:

1.) The surface sterilization procedure appears to be inefficient or not effective, leading to increases of some taxa. Reviewer 1 suggests measuring volatiles in parasitized caterpillars before and after surface-sterilization.

2.) The fact that the hyperparasitoid assay shows that the hyperparasitoid does not discriminate between parasitized and unparasitized caterpillars is problematic given the title of the paper and sections of the discussion. Perhaps this can be addressed and or clarified, but it does seem to diminish the significance of the present study, given that the authors have already published the effects of parasitism on the microbiome of P. brassicae (Gloder et al., 2021)

3.) It is pointed out that several conclusions are not currently supported by the data.

4.) The hypothesis that Wolbachia sp. Is the driver of mVOC changes in parasitized caterpillars should be explored in more detail.

We cannot make any decision about publication until we have seen the revised manuscript and your response to the reviewers' comments. Your revised manuscript is also likely to be sent to reviewers for further evaluation.

Sincerely,

Adler R. Dillman, Ph.D.

Guest Editor

PLOS Pathogens

P'ng Loke

Section Editor

PLOS Pathogens

Kasturi Haldar

Editor-in-Chief

PLOS Pathogens

orcid.org/0000-0001-5065-158X

Michael Malim

Editor-in-Chief

PLOS Pathogens

orcid.org/0000-0002-7699-2064

Based upon my own reading of the manuscript and the consensus of the three reviewers, this is a very interesting and well-written study of the odors involved in parasitism by hyperparasitoids. The attempts of the authors to investigate the origin of odors that are attractive to hyperparasitoids was appreciated. The research topic is fascinating.

I agree with the reviewers that some substantial revisions are necessary. In the aggregate, the three reviewers outline several major issues that the authors will need to address, and I agree with these. I consider that the most crucial of these are:

1.) The surface sterilization procedure appears to be inefficient or not effective, leading to increases of some taxa. Reviewer 1 suggests measuring volatiles in parasitized caterpillars before and after surface-sterilization.

2.) The fact that the hyperparasitoid assay shows that the hyperparasitoid does not discriminate between parasitized and unparasitized caterpillars is problematic given the title of the paper and sections of the discussion. Perhaps this can be addressed and or clarified, but it does seem to diminish the significance of the present study, given that the authors have already published the effects of parasitism on the microbiome of P. brassicae (Gloder et al., 2021)

3.) It is pointed out that several conclusions are not currently supported by the data.

4.) The hypothesis that Wolbachia sp. Is the driver of mVOC changes in parasitized caterpillars should be explored in more detail.

Reviewer's Responses to Questions

**Part I - Summary**

Reviewer #1: Bourne et al investigated how parasitic wasp affect the emitted volatiles and the microbiome of their host, and how these changes impact the foraging behavior of hyperparasitoids. Through the characterization of insect microbiomes, volatile measurements, and behavioral experiments, and different statistical analyses, the authors conclude that that parasitism by C. glomerata leads to a significant alteration of the odours of its host caterpillar P. brassicae and that the observed odour changes are related to changes in the internal and external microbiome of the caterpillars, and in turn determine hyperparasitoid preferences.

I find that the research topic is fascinating, but have major concerns as described below:

1. The authors conclude (Line 343) that: “In this study we show that parasitism by C. glomerata leads to a significant alteration of the odours of its host caterpillar P. brassicae. This statement is not supported by the data. Only three volatiles, (Z)-4-Hepten-1-ol, 3-Octanol, and 3-Octanone are significantly, and consistently induced upon parasitization. However, several compounds seem to be suppressed, but there is no statistical support for this statement nor these changes are always consistent across treatments.

2. The authors conclude (Line 344) that: The odour changes corresponded with changes in the internal and external microbiome of the caterpillars. This statement is not supported by the data. To fully support this statement, authors should measure volatiles in parasitized caterpillars before and after surface-sterilization. The authors conducted some of these measurements, although with starved larvae, which limits the interpretation of the results. Moreover, the surface sterilization procedure seems to be inefficient as there are some bacteria that increase (zOTU1 and zOTU2, some of the most abundant ones) after surface sterilization (Fig. 3). Apart from this, there is again no statistical support for this statement. There are significant changes across treatments, especially starvation has a drastic effect on volatile emissions, but volatile levels emitted by control and surface sterilized insects are not statistically different regardless of their starvation or parasitization status (Table 1). What the authors can say about this volatile data is that starvation drastically impact volatile emissions by the caterpillars.

3. Lastly, the authors conclude (Line 345) that the odour changes corresponded with …attraction of hyperparasitoids to parasitized caterpillars. Again, this statement is not supported by the data. Hyperparasitoids do not show any preference for parasitized over unparasitised caterpillars independently of surface sterilization treatments (See the results of the choice experiments between Pb-ST+W vs Cg-ST+W, and also between Pb-ST vs Cg-ST). What the authors can show is that hyperparasitoids preferred the odours of starved, parasitised larvae (Cg-ST) over odours from surface sterilized caterpillars (Cg-ST+W)(Fig 5), pointing to a potential role of bacteria (or their volatiles). However, there are no changes in volatile profiles between control and surface sterilized caterpillars (Table 1, discussed above in point 2). In addition, the surface sterilization treatments are not efficient, as discussed in above in point 2, and some bacteria even increase (zOTU1). Taken together, there seems to be additional factors (independently of microbial volatiles) that differ between surface sterilized, starved, parasitized caterpillars (Cg-ST+W) and non-sterilized, starved, parasitized caterpillars (Cg-ST) that can explain the preference of hyperparasitoids.

Minor comments:

1. Table 1. It is very difficult to actually see what are the volatiles that differ across treatments. I suggest that this table is moved to supplementary material and that the authors create bar plots for the volatiles that are statistically different across treatments, specially those that are relevant to interpret the results of the choice experiments.

2. The last paragraph of the introduction should be shorten and should be focused more on a short summary of the hypothesis being tested, a very brief summary of the main findings and put them in a broader context. Right now, the authors describe too many detailed results.

3. In the summary. I would write “The mVOCs might thereby provide a reliable cue to carnivorous enemies in locating their host or prey. “Parasitism by parasitoid wasps might alter the microbiome of their caterpillar host, affecting organismal odours and interactions with insects of higher trophic levels”.

Reviewer #2: In this manuscript, ‘Parasitism causes changes in caterpillar odours and associated bacterial communities with consequences for host-location by a hyperparasitoid’, the authors present a combination of chemical, microbial, and behavioral experiments to address the question of whether parasitism of Pieris brassicae caterpillars by the endoparasitoid Cotesia glomerata (or Hyposoter ebeninus) affects the microbial community, odor, and attractiveness of caterpillars to the hyperparasitoid Baryscapus galactopus. The authors describe previous work in a related caterpillar (Pieris rapae) that detected changes in caterpillar body odor that allow hyperparasitoids to located parasitized caterpillars, and work in Pieris brassicae (the focal species in this study) demonstrating changes in the internal microbiome after being parasitized. This manuscript aims to address whether odors that attract hyperparasitoids are produced by the caterpillar, caterpillar frass, or are microbial in origin (internal/external microbiome). This is a very interesting manuscript that represents a significant effort (particularly the sample sizes for the y-tube assay), however I did note several areas of concern.

Reviewer #3: This paper deals with the role of microorganisms as drivers of interaction between insects at multiple trophic levels. Specifically, evidence is provided that parasitism of the large cabbage white caterpillar Pieris brassicae by endoparasitoid wasp Cotesia glomerata led to the production of four characteristic volatile products and significantly affected the internal and external microbiome of the caterpillar, which were both found to have a significant correlation with caterpillar odors, with consequences for host-location by the hyperparasitoid Baryscapus galactopus. It was hypothesized that the changes in external microbiome and body odour after parasitism were driven by the resident internal microbiome of caterpillars, where the bacterium Wolbachia sp. was only present after parasitism.

The paper is well written and analyzes the role of the microbiota in a very interesting biological host-parasite interaction system. The study, however, appears to be mainly observational and also contains some incongruences. This unfortunately diminishes the enthusiasm in reading it. In particular, the hypothesized involvement of Wolbachia as a main driver in the system (as it results from forward selection redundancy analyses), is attractive, but deserves further investigation from a mechanistic point of view to strengthen the paper. Therefore, while containing elements of originality and novelty, the study appears, in my opinion, preliminary in its current state.

**Part II – Major Issues: Key Experiments Required for Acceptance**

Reviewer #1: (No Response)

Reviewer #2: There are two major concerns related to the results presented in this manuscript:

1) Statistical analysis of volatile data – The authors state (lines 582 through 585), “Supervised orthogonal partial least squares-discriminant analysis (OPLS-DA) and unsupervised principal component analysis (PCA) statistical models were used as a tool to compare and correlate treatment groups. Whereas PCA was used when no significant model for OPLS-DA could be obtained.”

For OPLS-DA the methods state that R2 and Q2 metrics were calculated and models were validated using permutations and CV-ANOVA. I do not see the model metrics and validation results reported in the manuscript. The variable importance scores (VIP) from OPLS-DA are the basis for many conclusions in this manuscript, but currently there is no information presented about the validity of the models.

Also related to analysis of volatile data, the results report p values from comparisons between treatments (e.g. line 161, line 165). Neither OPLS-DA nor PCA is a statistical test that produces p-values, and the methods do not appear to list other analyses that were conducted on the volatile data. It is not clear to me what test generated these results. Please clarify this, and any areas where p-values or other metrics are reported without indicating origin.

2) The bioassay results indicate that hyperparasitoid females did not exhibit any preference for parasitized caterpillars when given the choice between parasitized and unparasitized caterpillars. These bioassays were conducted a second time using another parasite species and the results were consistent. The outcome of these bioassays is summarized in lines 329-331, “The consistency of these results indicate that the preference of the hyperparasitoid is corelated with the presence of the external microbiome, though it is not discriminating between the odours of parasitised and unparasitised caterpillars in this setup.”

However, the discussion opening (line 343-346) states that, “…C. glomerata leads to a significant alteration of the odours of its host caterpillar P. brassicae. The odour changes corresponded with changes in the internal and external microbiome of the caterpillars and attraction of hyperparasitoids to parasitized caterpillars.”

Given these results, I do not understand logic behind this statement in the discussion. The hyperparasitoids appear to prefer an intact external microbiome regardless of of parisitism status, and direct comparisons testing the effect of parasitism status (Pb-ST vs. Cg-St and Pb-ST-W vs. Cg-ST-W) were non-significant.

This confusion also occurs when reading lines 428-430 versus lines 440-441.

Line 428-430: “…our results support the hypothesis that the changes after parasitism in the microbiome and caterpillar odour provide a mechanism by which hyperparsitoids can locate or distinguish hosts.”

Line 440-441: “It is remarkable that the hyperparasitoid B. galactopus is unable to discriminate between starved parasitised and unparasitised P. brassicae caterpillars based on their odours.”

Without more detailed explanation, the discussion appears contradictory. Please clarify these areas and provide more details related to the interpretation of results.

Reviewer #3: 1. Microbiome manipulation treatment. "Surface sterilization" (Ln. 520-523) has not been microbiologically verified. Gentle treatment of caterpillars (three washes with 0.05% sodium hypochlorite) is unlikely to be sufficient to "sterilize". Such a treatment could be sufficient to modify the relative abundance of some microorganisms as a consequence of the different sensitivity to sodium hypochlorite and adhesive capacity. Thus, the “decline of Staphylococcus sp. (zOTU 3) and increase of Enterococcus sp. (zOTU 1) relative abundances” (ln. 236) could be simply due to this effect. As a consequence, the suggestion that “the internal microbiome potentially acted as a source for the external microbiome” (242-243) is probable but not supported, also because Enterococcus sp. (zOTU 1) and Enterobacteriaceae sp. (zOTU 2) appear to be present and abundant in the external bacterial community even without surface sterilization treatment (Fig. 3d).

2. The result of forward selection redundancy analyses (RDA) (Fig. 4) is rather weak. A robust correlation between microorganisms and odors appears to be missing, with the exception of Wolbachia. However, this correlation is obvious, given that Wolbachia is transferred upon oviposition from the parasitoid into the caterpillar, and therefore its presence is intrinsically associated with C. glomerata parasitism. Probably the only relevant result is the reduction of Sphingomonas in the parasitized caterpillars, which should be better explored in this study.

3. The hypothesis that “Wolbachia sp. is the strongest candidate to cause the production of characteristic mVOCs for parasitised caterpillars” (ln. 457-460) is attractive, but should be investigated in more detail to strengthen the paper. One proposed mechanism is that Wolbachia sp. produces the four mVOCs related to parasitism directly by itself (ln. 417-418; ln. 460). Is there any evidence (even from genome analysis) that this is possible? Wolbachia has an obligate intracellular lifestyle that makes it difficult to work with this bacterium to address this question experimentally. However, as the authors discuss (ln. 423-424), insect cell-lines could provide a solution to characterize the presumed Wolbachia’s volatiles. The other two mechanisms proposed (ln. 418-421) also deserve further experimentation.

**Part III – Minor Issues: Editorial and Data Presentation Modifications**

Reviewer #1: (No Response)

Reviewer #2: 1) The introduction includes information that occasionally appears extraneous or review-like (beyond necessary context/background). It would be helpful to make the connections to this study clearer.

2) Line 75: “…Microorganisms can affect insect olfaction and behavior”

I do not disagree with this statement, however the potential for microorganisms to affect the molecular processes of olfaction is different from microorganisms’ affect on olfactory guided behavior.

3) Line 79: Are the parentheses necessary? “(body) odours”

4) Line 86-88: “Their preference is strongly linked to individuals with a higher bacterial abundance…”

This is a broad statement and both supporting examples related to human skin microbiota and mosquitoes. Is there literature that supports this in relation to carnivorous/hematophagous insects in general?

5) Figure 1 – If the primary goal of figure 1 is to outline the experimental design and sampling, I do not think the description of the aluminum foil restriction device is necessary here. It was described more clearly in the methods section.

6) Line 147 – the abbreviations for experimental treatments are described here for starved (ST) and surface sterilized caterpillars (W) but the abbreviations for unparasitized (Pb) and parasitized (Cg) are not defined.

7) Lines 161, 168, and line 7 of Table 1. Line 161 lists (Z)-3-hepten-1-ol as characteristic of parasitized caterpillars and refers to yellow highlighted lines of Table 1. Table 1 highlights (Z)-4-hepten-1-ol. Which compound name is correct? This also applies to line 168 which refers to Table A in the supporting information and line 358.

8) Line 181: What is the citation [44] included in support of? Methods?

9) Line 182-184: It seems appropriate to put these figures (S5 Fig) in the supplementary materials, however the statistical analyses should appear alongside the interpretation.

10) Table 1: The methods section indicates that these compound IDs represent tentative identification (methods lines 575-578). The figure legend should indicate that these are tentative and not authenticated identifications. Alternatively, many of these are readily available as pure standards and the IDs could be authenticated.

11) Also Table 1: The figure legend (and relevant methods) indicate that compound abundance was determined by peak height rather than peak area. This seems to assume that peaks are reliably gaussian. Is there a reason to use peak height rather than peak area?

12) Line 235: “parasitised, surface sterilized caterpillars (Cg-ST)”. Should this be ‘Cg-ST-W’?

13) Line 326: “contrasting comparisons of their microbiome, odour profiles (S3 and S4 Figs) and earlier findings [44].” This reference is for a separate species of caterpillar, which should be indicated (e.g. “…earlier findings in a closely related species” or similar). It is easy to confuse this as an earlier study of the same species.

14) Line 378-379: “…confirming their origin and indicating that frass odour is more dependent on food plant than caterpillar species.” Presumably the two caterpillar species were reared on the same host plant? This is not indicated.

15) Lines 384-386: Please make it clear that being a known/possible mVOC does not mean that a compound cannot appear in non-microbial contexts. Many of the compounds discussed here are reported in a variety of taxa.

16) Line 663: Permanova analyses are frequently accompanied by analyses of dispersion, as permanova alone does not distinguish between differences in centroid versus dispersion. Were tests done to distinguish between differences between multivariate means/centroids and differences in spread/variance?

17) Line 1039: Mann-Withney U-test should be Mann-Whitney.

Reviewer #3: 1. The indistinct use of the words "microbiome" and "microbiota" is incorrect. See the sentence in line 65: “The microorganismal part of the holobiont also called microbiome or microbiota”, but also check throughout the text.

2. Some results seem to partially inconsistent with those previously published, and the inconsistences are not sufficiently discussed. See the title of the ref. 47: “Parasitism by endoparasitoid wasps alters the internal but not the external microbiome in host caterpillars.”, in contrast with what is reported in the abstract of the present paper “We found that parasitism by C. glomerata … significantly affected the internal and external microbiome of the caterpillar, which were both found to have a significant correlation with caterpillar odours”, and also ln. 245-252.

PLOS authors have the option to publish the peer review history of their article (what does this mean?). If published, this will include your full peer review and any attached files.

Reviewer #1: No

Reviewer #2: No

Reviewer #3: No
---

## [Editor Report · Decision Letter 1]

2 Mar 2023

Dear Bourne,

We are pleased to inform you that your manuscript 'Parasitism causes changes in caterpillar odours and associated bacterial communities with consequences for host-location by a hyperparasitoid' has been provisionally accepted for publication in PLOS Pathogens.

Best regards,

Adler R. Dillman, Ph.D.

Guest Editor

PLOS Pathogens

P'ng Loke

Section Editor

PLOS Pathogens

Kasturi Haldar

Editor-in-Chief

PLOS Pathogens

orcid.org/0000-0001-5065-158X

Michael Malim

Editor-in-Chief

PLOS Pathogens

orcid.org/0000-0002-7699-2064

The authors have done an extensive re-write of the paper, incorporating the reviewers’ concerns and suggestions for improvement in detail. This includes due attention to the many suggested edits and improvements highlighted by Reviewers collectively. Additional experiments were performed to address concerns about hyperparasitoid assays. Conclusions have been tempered and more carefully written.

The hypothesis that Wolbachia sp. drives mVOC changes was also more thoroughly explored. I appreciate the extensive and detailed work done by the authors to address reviewer comments and suggestions. I congratulate them on their excellent work!

I see no shortcomings in the revised submission and no reason to delay publication further with additional review.
---

## [Editor Report · Acceptance letter]

17 Mar 2023

Dear Bourne,

We are delighted to inform you that your manuscript, "Parasitism causes changes in caterpillar odours and associated bacterial communities with consequences for host-location by a hyperparasitoid," has been formally accepted for publication in PLOS Pathogens.

Best regards,

Kasturi Haldar

Editor-in-Chief

PLOS Pathogens

orcid.org/0000-0001-5065-158X

Michael Malim

Editor-in-Chief

PLOS Pathogens

orcid.org/0000-0002-7699-2064